# Score-based generative models are provably robust: an uncertainty quantification perspective

**Nikiforos Mimikos-Stamatopoulos**
Department of Mathematics
Université Côte d'Azur
nmimikos@unice.fr

**Benjamin J. Zhang**
Division of Applied Mathematics
Brown University
benjamin_zhang@brown.edu

**Markos A. Katsoulakis**
Department of Mathematics and Statistics
University of Massachusetts Amherst
markos@umass.edu

## Abstract

Through an uncertainty quantification (UQ) perspective, we show that score-based generative models (SGMs) are provably robust to the multiple sources of error in practical implementation. Our primary tool is the Wasserstein uncertainty propagation (WUP) theorem, a *model-form UQ* bound that describes how the $L^2$ error from learning the score function propagates to a Wasserstein-1 ($\mathbf{d}_1$) ball around the true data distribution under the evolution of the Fokker-Planck equation. We show how errors due to (a) finite sample approximation, (b) early stopping, (c) score-matching objective choice, (d) score function parametrization expressiveness, and (e) reference distribution choice, impact the quality of the generative model in terms of a $\mathbf{d}_1$ bound of computable quantities. The WUP theorem relies on Bernstein estimates for Hamilton-Jacobi-Bellman partial differential equations (PDE) and the regularizing properties of diffusion processes. Specifically, *PDE regularity theory* shows that *stochasticity* is the key mechanism ensuring SGM algorithms are provably robust. The WUP theorem applies to integral probability metrics beyond $\mathbf{d}_1$, such as the total variation distance and the maximum mean discrepancy. Sample complexity and generalization bounds in $\mathbf{d}_1$ follow directly from the WUP theorem. Our approach requires minimal assumptions, is agnostic to the manifold hypothesis and avoids absolute continuity assumptions for the target distribution. Additionally, our results clarify the *trade-offs* among multiple error sources in SGMs.

## 1 Introduction

Score-based generative models (SGMs) [1, 2] are highly effective [3], producing high quality samples, with more stable and less computationally intensive training methods than generative adversarial nets and normalizing flows [4]. The models are empirically robust to approximations and errors in learning the score function. While SGM generalization properties have been studied for idealized conditions [5, 6, 7, 8], analyses of their robustness in practical settings remains underexplored. This paper analyzes SGMs through the *regularity theory of nonlinear partial differential equations* (PDEs) [9], specifically Hamilton-Jacobi-Bellman (HJB) equations [10]. Our main result is the *Wasserstein uncertainty propagation* (WUP) theorem (Theorem 3.1), a versatile model-form uncertainty quantification (UQ) bound which we use to theoretically explain the robustness of SGMs to approximation errors in

38th Conference on Neural Information Processing Systems (NeurIPS 2024).

practical implementation. Generalization bounds for integral probability metrics (IPMs), such as the Wasserstein-1 ($\mathbf{d}_1$) and the total variation (TV) distance follow directly from the WUP theorem.

In the context of SGMs, the WUP theorem shows how an $L^2$ neighborhood around the true score function propagates to an IPM neighborhood around the true data distribution. By relating how various approximations in SGMs contributes to $L^2$ error with respect to the true score function, we establish how well the resulting SGM generalizes. Theorem 3.2 shows how the error in the learned score-function with respect to the explicit score-matching objective, or uniform-in-time $L^2$ error, and the choice of initial condition define the radii of $\mathbf{d}_1$ and TV neighborhoods, respectively. Additionally, Theorem 3.3 addresses the case where the score function is learned from finite data using the denoising score-matching (DSM) objective and incorporates early stopping.

Our bounds capture *trade-offs* among the errors. The ability of SGMs to generalize depends on choices such as the early stopping time and how overtraining to the DSM objective and neglecting early stopping lead to SGMs that *memorize* and overfit to the data. Notably, our approach relies on minimal assumptions on the data distribution, being agnostic to the manifold hypothesis. This contrasts with existing convergence results which assume the hypothesis [8] or not [5, 7] *a priori*. Furthermore, unlike previous work, we obtain our generalization bounds with respect to the $\mathbf{d}_1$ distance directly, *without* appealing to the Girsanov theorem, the Kullback-Leibler divergence, the $\chi^2$ divergence, or Pinsker's inequality [5, 11, 7]. This suggests our bounds may be *sharper* than those which bound stronger norms and divergences. A notable feature of our DSM generalization bound is that the error bound is a computable function of the DSM objective, contrasting with prior results typically assume the learned score is close to the truth with respect to the ESM objective [5].

The WUP theorem also enables *robust uncertainty quantification* (UQ) for score-based generative models, a rare capability in generative models. Robust UQ recognizes that learning complex models involves multiple sources of uncertainty due to modeling choices and imperfect data. The *distributionally robust* perspective [12] quantifies a uncertainty set based on divergences and probability metrics [13, 14, 15, 16] to measure the impact of model uncertainty around a baseline model. The drawback is that these sets are typically difficult to find computationally. The WUP theorem is an example of a robust UQ bound for IPMs that is computable.

## 1.1 Contributions

- We introduce the use of *regularity theory of nonlinear PDEs* for analyzing generative flows [9, 10]. Our key idea is using the *Kolmogorov backward equation* to study the evolution of IPMs under a generative flow, yielding generalization bounds. WUP is one particular application of this idea, and while we use it to study SGMs, this analysis is not limited to only SGMs. We show that the regularizing properties of the underlying Fokker-Planck equation imply SGM's robustness to errors with few assumptions on the data distribution. An intuitive explanation of our main technical contribution is provided in Section 4.

- The WUP theorem (Theorem 3.1), which we state and prove, describes how an $L^2$ ball centered around the true score function maps to a neighborhood of IPMs (such as $\mathbf{d}_1$, TV, and MMD [17]). WUP is a *model-form uncertainty quantification bound*, which maps uncertainties introduced by modeling choices when practically implementing SGM. The resulting generalization bounds WUP produces demonstrate that with properly chosen model parameters, SGM is robust to errors due to score-matching, finite sample approximation, early stopping time, and choice of the reference distribution. (Theorems 3.2, 3.3, and C.1).

- When applied to SGM, WUP produces generalization bounds under minimal hypotheses on the data distribution and score function, and explains the impact of implementation errors has on generalization. Our approach is *agnostic* to the manifold hypothesis, applying whether or not the data distribution has a density. Moreover, it is adaptable to additional assumptions about the target distribution to yield improved generalization bounds.

## 1.2 Related work

Convergence and generalization of SGMs have been well-studied. Many approaches [5, 7, 18] assume the target distribution is absolutely continuous with respect to a Gaussian, and obtain generalization bounds for TV, $\chi^2$, and $\mathbf{d}_1$ by bounding the KL divergence, a stronger divergence, via the Girsanov theorem [11, 6, 18], Pinsker's inequality [5], functional inequalities [7], or other means [11, 19].

While [20] shows that kernel estimators for the score function are minimax optimal distribution estimators, they make no connections to deep learning-based SGMs, and they assume the target distribution is sub-Gaussian and is in a Sobolev space. Meanwhile, [6] has comprehensive sample complexity results that considers neural net approximations in the finite sample regime. Their results, however, are specialized to distributions in a Besov space, and, similar to other work [5, 11, 7, 19], rely on bounding the KL divergence, which breakdown under the manifold hypothesis.

Our results are similar to [8] which derives $\mathbf{d}_1$ bounds directly, proving SGM convergence under the manifold hypothesis. However, the analysis assumes a particular discretization of the SGM. In [21], an uncertainty propagation bound for the Wasserstein-2 distance is proven using a similar approach to this work. Their work, however, relies on strong, uncheckable assumptions on the score function and target measure, does not address the errors we study here, and is not extendable to IPMs.

## 2 Background and notation

Let dimension $d \in \mathbb{N}$, and terminal time $T > 0$. Denote $\mathbb{T}^d \subset \mathbb{R}^d$ to be the unit torus and $R\mathbb{T}^d \subset \mathbb{R}^d$ to be the torus of radius $R > 0$. Let $\mathcal{P}(\Omega)$ be the space of probability distributions on $\Omega$, where $\Omega$ is $\mathbb{R}^d$ or $R\mathbb{T}^d$. Let $\pi \in \mathcal{P}(\Omega)$ be the target data distribution, known only through a finite number of samples $\{z_j\}_{j=1}^N \sim \pi$. Let $f : [0, T] \times \mathbb{R}^d \to \mathbb{R}$ be a vector field and $\sigma : \mathbb{R} \to \mathbb{R}$ be a positive function. Score-based generative modeling [1, 2] is based on considering two Stochastic Differential Equations (SDEs for short) whose flow maps are inverses of each other. Define $Y(s)$, $X(t)$ to be the forward and reverse diffusion processes over time interval $s, t \in [0, T]$ that evolve according to

$$\mathrm{d}Y(s) = -f(T - s, Y(s))\mathrm{d}s + \sigma(T - s)\mathrm{d}W(s), \quad Y(0) \sim \pi \tag{1}$$

$$\mathrm{d}X(t) = \left[ f(t, X(t)) + \sigma(t)^2 \nabla \log \eta^\pi (T - t, X(t)) \right] \mathrm{d}t + \sigma(t)\mathrm{d}W(t), \quad X(0) \sim m_0, \tag{2}$$

where $Y(s) \sim \eta^\pi(s, \cdot)$, and $W(t)$ is a standard Brownian motion in $\mathbb{R}^d$. Heres, $\eta^\pi(s, \cdot)$ is the density of $Y(s)$ at time $s$ where the initial distribution was $\pi$. From [22], it is known that if $m_0 = \eta^\pi(T, \cdot)$, then $X(t) \sim \eta^\pi(T - t, \cdot)$. SGMs generated new samples from $\pi$ simulating trajectories of the reverse process (2) using an approximate score function $\mathbf{s}_\theta \approx \mathbf{s} = \nabla \log \eta^\pi$. Typically $f$ and $\sigma$ are chosen such that $\eta^\pi(T, \cdot)$ is well-approximated by a normal distribution, which is then used as the initial distribution.

The score function is learned via samples $\{z_j\}_{j=1}^N$ from $\pi$ by minimizing one of several *score-matching* objectives. Let $\mathbf{s}_\theta$ be some function where the parameters $\theta$ are learned via one of three score-matching objectives. For a path $\rho : [0, T] \to \mathcal{P}(\Omega)$, we define the functionals

$$\mathcal{J}(\rho, \theta) = \int_0^T \int_\Omega |\mathbf{s}_\theta - \nabla \log \rho|^2 \, \mathrm{d}\rho(s)\mathrm{d}s \quad \mathcal{J}_L(\rho, \theta) = \int_0^T \int_\Omega \left( |\mathbf{s}_\theta|^2 + 2\nabla \cdot \mathbf{s}_\theta \right) \mathrm{d}\rho(s)\mathrm{d}s, \tag{3}$$

where the subscript $L$ highlights the linear dependence of $\mathcal{J}_L$ on the underlying measure $\rho$. The *explicit score matching objective* (ESM) is $J_{\mathrm{ESM}}(\eta^\pi, \theta) = \mathcal{J}(\eta^\pi, \theta)$. As evaluation of the true score function $\nabla \log \eta^\pi$ is typically inaccessible, the *implicit* (ISM) $J_{\mathrm{ISM}}(\eta^\pi, \theta) = \mathcal{J}_L(\eta^\pi, \theta)$ [23] or the *denoising* (DSM) *score-matching* objectives [24, 1] are computed in practice

$$J_{\mathrm{DSM}}(\eta^\pi, \theta) = \int_0^T \int_\Omega \int_\Omega \left| \mathbf{s}_\theta - \nabla \log \eta^{x'} \right|^2 \mathrm{d}\eta^{x'}(s)\mathrm{d}\pi(x')\mathrm{d}s.$$

Here $\eta^{x'}(s)$ denotes the probability transition kernel from $x'$ to $x$ of (1) at time $s$. The DSM objective is most frequently used in practice as it does not require computing derivatives of $\mathbf{s}_\theta$. It does, however, require the evaluation of $\eta^{x'}(s)$ in closed form, which is typically only accessible for linear SDEs.

**Remark 2.1** (Choice of domain $\Omega$). *While our approach will also produce bounds when $\Omega = \mathbb{R}^d$, we primarily focus on the torus $R\mathbb{T}^d$, which is equivalent to a bounded domain with periodic boundary conditions. This choice ensures that the long-time behavior of the noising process (1) converges to the uniform distribution on $R\mathbb{T}^d$. Therefore, we set $f = \mathbf{0}$ and $\sigma(t) = \sqrt{2}$ instead of using the Ornstein-Uhlenbeck process. We make this choice for mathematical clarity, however our results generally apply, with minor modifications, to the entire space $\mathbb{R}^d$ and for other noising processes.*

### 2.1 Equivalence of score-matching objectives in the finite sample regime

Finite sample approximations of ESM, ISM, and DSM are not generally equivalent. However, [24, 25], show that DSM becomes equivalent to ESM and ISM in the finite sample regime when

$\eta^\pi(s)$ is replaced with its *kernel* density estimate $\eta^N(s)$, where $\eta^N(0) = \pi^N = \frac{1}{N}\sum_{i=1}^N \delta_{z_i}$. The kernel estimate, however, does not have a well-defined score at $s = 0$, so the DSM objective is often integrated only for $s \in [\epsilon, T]$, an example of early-stopping in SGM [1, 26]. In continuous time, this has the effect of score-matching for the mollified distribution $\pi^{N,\epsilon} = \pi^N \star \rho_\epsilon$, where $\rho_\epsilon$ is the transition probability kernel for $s = \epsilon$[1]. Then it can be shown that

$$\mathcal{J}(\eta^{N,\epsilon}, \theta) = J_{\text{DSM}}(\eta^{N,\epsilon}, \theta) = J_{\text{ESM}}(\eta^{N,\epsilon}, \theta) = J_{\text{ISM}}(\eta^{N,\epsilon}, \theta) + 4\|\nabla\sqrt{\eta^{N,\epsilon}}\|_2^2. \tag{4}$$

## 3 An uncertainty quantification approach to generalization in SGMs

We describe our UQ approach to generalization in SGMs and overview our main results. Discussion of the proof methods is deferred to Sections 4 and 5. Our primary goal is to study how practical and approximation errors made in implementing SGMs translate into errors in the generative distribution relative to the true distribution.

### 3.1 Source of errors in score-based generative modeling

We attribute errors of SGM to the following six sources:

- **Data distribution is only accessible via samples** $e_1$: The target distribution is only known through a finite set of samples. In practice, the regularity of the score function and data distribution, such as Lipschitzianity or whether the distribution has a density, is unknowable.

- **Choice of score-matching objective** $e_2$: In practice, score-matching objectives are approximated via samples. The DSM objective is most frequently used as it avoids computing derivatives of the score function. While DSM and ISM are equivalent given the exact density evolution $\eta(x, s)$, they differ when approximated with finite samples. Previous analysis typically assumes that the learned score function is close to the true score function within some $L^2$ distance, i.e., a ball determined by $J_{\text{ESM}}$. In contrast to previous work [11, 7, 5], we show in Theorem B.5 how training through DSM translates into a bound for ESM. Moreover we prove Proposition B.8 which states that if there exists a neural net that well-approximates the true score then that the target density is necessarily regular.

- **Expressiveness of the score function** $e_3$. The expressivity of neural net approximations to the score function will depend on the particular expressivity of the parametrization.

- **Choice of reference distribution** $e_4$. With access to the exact score function, the denoising process (2) produces trajectories that sample from $\pi$ only if the initial condition starts at $\eta^\pi(\cdot, T)$. In practice, however, the initial distribution is usually a Gaussian approximation of $\eta^\pi$ or, in the case of the OU noising process, its corresponding stationary distribution.

- **Early stopping of the denoising process** $e_5$. In practice, the score-matching objective is integrated from $s \in [\epsilon, T]$, for small $\epsilon$, instead of $s = 0$ [1, 26]. This prevents the SGM from *memorizing* the training data [27, 28], and is equivalent to running the denoising process for $t \in [0, T - \epsilon]$. Early stopping is *crucial* when the data distribution is supported on a low-dimensional manifold, where it has no density with respect to Lebesgue measure. Previous analyses of SGM often adjust $\epsilon$ to optimize generalization bounds [6, 20].

- **Discretization error** $e_6$. The denoising SDE must be solved via a numerical scheme. Previous work [5, 11, 7, 8] have considered the impact of discretization error on the generalization abilities of SGM. While our analysis does not consider the discretization error, our approach can be extended to study discretization error through the use of modified equations [29].

### 3.2 Model-form uncertainty quantification

Let $\mathbf{d}$ be an *integral probability metric* (IPM) between measures $\nu_1, \nu_2 \in \mathcal{P}(\Omega)$

$$\mathbf{d}(\nu_1, \nu_2) = \sup_{\psi \in \mathcal{X}} \int \psi(x) d(\nu_2 - \nu_1), \tag{5}$$

---

[1] Here, $\star$ denotes convolution.

for some function space $\mathcal{X}$. We assume $\mathcal{X} = \{\psi : \Omega \to \mathbb{R}, \|\nabla\psi\|_\infty \leq 1\}$ or $\{\psi : \Omega \to \mathbb{R}, \|\psi\|_\infty \leq 1\}$, corresponding to the Wasserstein-1 ($\mathbf{d}_1$) and total variation distances, respectively. Recall that if $\nu_1, \nu_2$ have densities, then their TV distance is equivalent to the $L^1$ distance between their densities.

Let $\pi$ be the target data distribution and take the stationary distribution of the noising process on the torus $m_g(0) = \frac{1}{vol(R\mathbb{T}^d)}$ as the initial condition. Given a learned score function $\mathbf{s}_\theta$, let $m_g(T)$ be the generative distribution produced by SGM. We study how IPMs between $\pi$ and $m_g(T)$ relate to errors from the first five sources of errors discussed above, i.e., we seek bounds of the form

$$\mathbf{d}(m_g(T), \pi) \leq \mathcal{F}(e_1, e_2, e_3, e_4, e_5). \tag{6}$$

Our key insight is that for score-based generative modeling, we can derive bounds of the form (6) via a *model-form uncertainty quantification* bound for the generative Fokker-Planck equation. The Wasserstein Uncertainty Propagation theorem formalizes this insight.

## 3.3 Wasserstein uncertainty propagation theorem

The WUP theorem is a general statement about how $L^2$ neighborhoods in the space of drift functions map to neighborhoods in $\mathcal{P}(\Omega)$ defined by IPMs.

**Theorem 3.1** (Wasserstein Uncertainty Propagation). *Let $\Omega = R\mathbb{T}^d$ or $\mathbb{R}^d$. Let $b^1, b^2 : [0, T] \times \Omega \to \mathbb{R}^d$ be given with $\|\nabla b^1\|_\infty < \infty$, and $m_1, m_2 \in \mathcal{P}(\Omega)$. If $m^i$ for $i = 1, 2$ are given by*

$$\partial_t m^i - \Delta m^i - \operatorname{div}(m^i b^i) = 0, \quad m^i(0) = m_i \tag{7}$$

*then, up to a universal constant $C > 0$, we have the following:*

- *If* $\displaystyle\sup_{0 \leq t \leq T} \|(b^2 - b^1)(t)\|_{L^2(m^2(t))} := \sup_{0 \leq t \leq T} \left( \int_\Omega \left|(b^2 - b^1)(t, x)\right|^2 m^2(t, x) dx \right)^{1/2} \leq \varepsilon_1,$
  *then*

$$\|m^2(T) - m^1(T)\|_{L^1(\Omega)} \leq C(\sqrt{T\|\nabla b^1\|_\infty} + 1) \left( \frac{1}{\sqrt{T}} \mathbf{d}_1(m_1, m_2) + \sqrt{T}\varepsilon_1 \right), \tag{8}$$

  *and*

$$\|m^2(T) - m^1(T)\|_{L^1(\Omega)} \leq C(\sqrt{T\|\nabla b^1\|_\infty} + 1) \left( \|m_1 - m_2\|_{L^1(\Omega)} + \sqrt{T}\varepsilon_1 \right). \tag{9}$$

- *For $\Omega = R\mathbb{T}^d$, if* $\|b^2 - b^1\|_{L^2(m^2)} := \left( \int_0^T \int_\Omega \left|(b^2 - b^1)(t, x)\right|^2 m^2(t, x) dx dt \right)^{1/2} \leq \varepsilon_2,$
  *then*

$$\mathbf{d}_1(m^2(T), m^1(T)) \leq CR^{\frac{3}{2}}(1 + \sqrt{\|\nabla b^1\|_\infty}) \left( \mathbf{d}_1(m_1, m_2) + \varepsilon_2 \right). \tag{10}$$

Equation (8) is a particularly notable result as we bound the TV distance between $m^1(T)$ and $m^2(T)$ in terms of a weaker $\mathbf{d}_1$ metric between $m_1$ and $m_2$. This is due to the regularizing effects of diffusion processes, which is showcased in the proof. See Section 4 and Section A.1 for full details.

## 3.4 Robustness of errors under ESM

We use WUP to produce generalization bounds when the only errors are due to the choice of the initial condition and the approximation of the score function with respect to the ESM objective.

**Theorem 3.2** (ESM bounds). *Let $\pi \in \mathcal{P}(\Omega)$ where $\Omega = R\mathbb{T}^d$ for some $R > 0$. Moreover let $e_{nn} > 0$. Assume that the learned score function $\mathbf{s}_\theta$ is such that $\mathcal{J}(\eta^\pi, \theta) \leq e_{nn}$. Then for $\mathbf{b}_\theta = \mathbf{s}_\theta(T - t, x)$ the generated distribution $m_g(T) \approx \pi$ where*

$$\partial_t m_g - \Delta m_g - \operatorname{div}(m_g \mathbf{b}_\theta) = 0 \text{ in } (0, T] \times \Omega, \quad m_g(0) = \frac{1}{vol(R\mathbb{T}^d)} \text{ in } \Omega, \tag{11}$$

*satisfies*

$$\mathbf{d}_1(\pi, m_g(T)) \leq CR^{\frac{3}{2}}(1 + \sqrt{\|\nabla \mathbf{s}_\theta\|_\infty}) \Big( \underbrace{Re^{-\frac{\omega T}{R^2}} \mathbf{d}_1\Big(\pi, \frac{1}{vol(R\mathbb{T}^d)}\Big)}_{\text{Error from choice of reference measure } (e_4)} + \underbrace{\sqrt{e_{nn}}}_{\text{Score function error } (e_3)} \Big).$$

$$\tag{12}$$

*If the stronger estimate* $\sup\limits_{0 \le t \le T} \|\mathbf{s}_\theta - \nabla \log(\eta^\pi)\|^2_{L^2(\eta^\pi(t))} \le e_{nn}$, *holds, then*

$$\|m_g(T) - \pi\|_{L^1(\Omega)} \le C(\sqrt{T}\|\nabla\mathbf{s}_\theta\|_\infty + 1)\left(\frac{R^2 e^{-\frac{\omega T}{R^2}}}{\sqrt{T}}\mathbf{d}_1\left(\pi, \frac{1}{vol(R\mathbb{T}^d)}\right) + \sqrt{T}e_{nn}\right). \quad (13)$$

*Here, the constants* $C, \omega$ *depend only on the dimension* $d$.

Applying WUP for $b^1$ and $b^2$ to be the true and learned score function, respectively, with Proposition D.3 on the convergence of the noising process to the stationary measure immediately yields the estimates above. Notice that the TV estimate (13) is comparable to Theorem 2 of [5], which also assumes a uniform-in-time $L^2$-accurate score function. Again, notice that we are able to bound the TV distance between $m_g(T)$ and $\pi$ using the weaker $\mathbf{d}_1$ distance between $\pi$ and $\frac{1}{\text{vol}(R\mathbb{T}^d)}$.

## 3.5 Robustness of errors under DSM

In practice, the score is learned through samples using the DSM objective with an early stopping parameter [1, 26]. SGM is effective at producing samples from distributions supported on (or near) low dimensional manifolds. Our $\mathbf{d}_1$ generalization bound describes how early stopping aids in generalization.

**Theorem 3.3.** *(Pointwise DSM generalization) Let* $\mathbf{b}_\theta = \mathbf{s}_\theta(T-t, x)$ *and* $m_g : [0,T] \times R\mathbb{T}^d \to \mathbb{R}$ *be given by* (11). *Assume the learned score function is such that* $J_{DSM}(\eta^{N,\epsilon}, \theta) = \mathcal{J}(\eta^{N,\epsilon}, \theta) < e_{nn}$. *Let* $0 < \delta < \epsilon$ *be such that* $\delta \le \hat{\pi}^{N,\epsilon}$, $\delta < \pi^\epsilon$. *Then up to a dimensional constant* $C = C(d) > 0$,

$$\mathbf{d}_1(\pi, m_g(T)) \lesssim \underbrace{\sqrt{\epsilon}}_{\text{Early stopping }(e_5)} + R^{3/2}(1 + \sqrt{\|\nabla\mathbf{s}_\theta\|_\infty})\bigg(\underbrace{Re^{-\frac{\omega T}{R^2}}\mathbf{d}_1\left(\pi, \frac{1}{vol(R\mathbb{T}^d)}\right)}_{\text{Choice of reference measure }(e_4)} + \sqrt{e'_{nn}}\bigg), \quad (14)$$

*where*

$$\sqrt{e'_{nn}} \lesssim \underbrace{\sqrt{e_{nn}}}_{\text{DSM score function error }(e_3)} + \underbrace{\sqrt{\left(1 + \frac{|\log(\delta)|}{\sqrt{\epsilon}} + T\|\mathbf{s}_\theta\|^2_{C^2([0,T]\times\Omega)}\right)\underbrace{\mathbf{d}_1(\pi^N, \pi)}_{\text{Finite sample error }(e_1)}}}_{\text{Choice of SM objective }(e_2)}. \quad (15)$$

The pointwise DSM generalization bound applies to every finite training sample of size $N$. A crucial part of this theorem relates the error in the practical DSM objective function to the ESM error needed to apply the WUP theorem. We connect the DSM objective early stopping to the ESM error with the mollified distribution $\pi^\epsilon$ in Lemma B.5. This result is *agnostic* to the manifold hypothesis for $\pi$, as long as $\epsilon > 0$. Such bounds will blow up if the KL divergence were used instead. In fact, previous results that use the KL divergence to bound the TV distance [5, 19, 11, 6] will produce vacuous bounds for the $\mathbf{d}_1$ distance under the manifold hypothesis as their $\mathbf{d}_1$ generalization bounds are derived by bounding the KL divergence.

**Remark 3.4** (Density lower bound). *Similar to [6], our DSM generalization bound relies on a density lower bound assumption. We note however, that our DSM generalization bound holds for any random sample* $\{z_i\}_{i=1}^N \sim \pi$. *This density lower bound assumption can be removed via Jensen's inequality if we consider the expected* $\mathbf{d}_1$ *distance between* $\pi$ *and* $m_g(T)$ *over random empirical measures. See Theorem C.1 and its proof in Section C.*

**Remark 3.5** (Trade-offs and memorization). *Theorem 3.3 captures trade-offs when training SGMs and memorization effects. To minimize the error from early stopping, we can let* $\epsilon \to 0$. *However, empirically, without early stopping, SGMs overfit to the kernel approximation and memorize the training data [27, 28, 25]. The bound is vacuous when* $\epsilon = 0$ *regardless of whether the distribution lies on a low-dimensional manifold. As* $\epsilon \to 0$, *training the DSM to be small implies that the score function must approximate a rough function with large Lipschitz constant, which will increase the bound. This shows that overtraining the DSM objective may not necessarily yield a better generative model. Moreover, suppose that* $\pi^N = \pi$, *then as* $\epsilon \to 0$ *and* $e_{nn} \to 0$, *we have that* $\mathbf{d}_1(\pi, m_g(T)) \to 0$, *indicating the model memorizes the training data. Our results corroborate those of [25].*

# 4 Regularity theory of Hamilton-Jacobi-Bellman PDEs enables uncertainty quantification in SGMs

A recent result in [30] established connections between generative flows with the theory of PDEs, more specifically the theory of mean field games. We continue investigating these connections and showcase how one may study generative modeling algorithms by obtaining stability estimates for the Fokker-Planck equation. We provide a proof sketch for our Wasserstein Uncertainty Quantification theorem (Theorem 3.1). Our strategy involves (1) constructing test functions for the IPMs using the Kolmogorov backward equation, (2) choosing the suitable function space for the terminal data depending on the desired IPM, and (3) obtaining gradient estimates of the test functions via Bernstein estimates. The theorem relies on the gradient of the test function remaining bounded for $t \in [0, T)$, which is guaranteed by the regularizing properties of diffusion processes. See A.1 for full details about the proof.

## 4.1 Kolmogorov backward equation determines suitable test functions

From (7), we aim to compute bounds for $\mathbf{d}(m^1(T), m^2(T)) = \sup_{\psi(x) \in \mathcal{X}} \int \psi(m^1(T) - m^2(T)) dx$. The measure $\lambda = m^1 - m^2$ satisfies the PDE

$$\partial_t \lambda - \Delta \lambda - \text{div}(\lambda b^1 + m^2(b^1 - b^2)) = 0 \text{ in } (0, T) \times \Omega, \quad \lambda(0) = m_2 - m_1 \text{ in } \Omega. \quad (16)$$

Let $\phi : [0, T] \times \Omega \to \mathbb{R}$ be a test function in space and time. We integrate in space and time against the PDE (16) and integrate by parts to move the derivatives on to $\phi$ which yields

$$\int_\Omega \lambda(T, x)\phi(T, x) - \lambda(0, x)\phi(0, x)dx + \int_0^T \int_\Omega \lambda \left(-\partial_t \phi - \Delta \phi + b^1 \cdot \nabla \phi\right) dxdt \quad (17)$$

$$+ \int_0^T \int_\Omega m^2 \nabla \phi \cdot (b^1 - b^2) dxdt = 0$$

Notice that if we choose the test function $\phi$ to satisfy the *Kolmogorov backward equation* (KBE)

$$-\partial_t \phi - \Delta \phi + b^1 \cdot \nabla \phi = 0 \text{ in } [0, T) \times \Omega, \quad \phi(T, x) = \psi(x) \text{ in } \Omega \quad (18)$$

with terminal condition $\psi \in \mathcal{X}$, then from (17), we have

$$\int_\Omega \lambda(T, x)\psi(x)dx = \int_\Omega \lambda(0, x)\phi(0, x)dx + \int_0^T \int_\Omega m^2(t)\nabla\phi(t, x) \cdot (b^2 - b^1)(t)dxdt. \quad (19)$$

The equality above is valid for any terminal condition $\psi$. Depending on the choice of function space $\mathcal{X}$ for $\psi$, we obtain bounds on different IPMs. Taking the supremum over $\mathcal{X}$ we have

$$\mathbf{d}(m^1(T), m^2(T)) \le \sup_{\psi \in \mathcal{X}} \left| \int_\Omega \lambda(0, x)\phi(0, x)dx \right| + \sup_{\psi \in \mathcal{X}} \left| \int_0^T \int_\Omega m^2 \nabla \phi \cdot (b^2 - b^1) dxdt \right|. \quad (20)$$

Recall that $\phi$ is related to $\psi$ via the KBE (18) To obtain bounds for $\mathbf{d}$, we need to bound the two terms in (20), which depend on the choice of function space $\mathcal{X}$ and assumptions on the drift terms.

## 4.2 IPM bounds depend on choice of terminal function space and gradient estimates

We split our proof sketch into two parts; the first part focuses on deriving $L^1$ estimates, while the second derives $\mathbf{d}_1$ estimates.

$L^1$ **estimates.** We first note that because of the *regularizing* properties of (18) we can obtain bounds on $\int_\Omega \lambda(0, x)\phi(0, x)dx$ with weaker norms on $\phi$. Notice that

$$\int_\Omega \lambda(0, x)\phi(0, x)dx \le \|\lambda(0)\|_{\mathcal{Y}'}\|\phi(0)\|_{\mathcal{Y}}, \quad (21)$$

where $\mathcal{Y}$ denotes a generic space of functions, and $\mathcal{Y}'$ is its dual. In what follows, we show that regularizing effects of (18) takes functions in $\mathcal{X}$ to functions with more regularity $\mathcal{Y}$ such that $\mathcal{Y}$ is compactly embedded in $\mathcal{X}$. This in turn implies that $\|\cdot\|_{\mathcal{Y}'}$ is weaker than $\mathbf{d}$ and so we are able to bound the stronger norm $\mathbf{d}$ by the weaker norm.

- To prove (8), we $\mathbf{d} = \|\cdot\|_{L^1(\Omega)}$, which corresponds with $\mathcal{X} = \{\psi : \Omega \to \mathbb{R}, \|\psi\|_\infty \le 1\}$. Observe that we can take $\mathcal{Y} = \{\psi : \Omega \to \mathbb{R}, \|\nabla\psi\|_\infty \le 1\}$, in which case we obtain

$$\int_\Omega \lambda(0, x)\phi(0, x)dx = \|\nabla\phi(0, x)\|_\infty \int_\Omega (m_1 - m_2)\frac{\phi(0, x)}{\|\nabla\phi(0, x)\|_\infty}dx \qquad (22)$$
$$\le \mathbf{d}_1(m_1, m_2)\|\nabla\phi(0, x)\|_\infty.$$

Notice that this bound crucially depends on showing that $\phi(0, x)$ is Lipschitz.

- To prove (9), we can simply take $\mathcal{X} = \mathcal{Y}$, and obtain

$$\int_\Omega \lambda(0, x)\phi(0, x) \le \|\lambda(0, x)\|_{L^1(\Omega)}\|\phi(0, x)\|_\infty. \qquad (23)$$

For (8) and (9), Cauchy-Schwarz on the spatial integral shows the second term can be bounded as

$$\int_0^T \int_\Omega m^2 \nabla\phi \cdot (b^2 - b^1)dxdt \le \int_0^T \|(b^1 - b^2)(t)\|_{L^2(m^2(t))}\|\nabla\phi(t, x)\|_{L^2(m^2(t))}dt \qquad (24)$$
$$\le \sup_{0 \le t \le T} \|(b^1 - b^2)(t)\|_{L^2(m^2(t))} \int_0^T \|\nabla\phi(t, \cdot)\|_\infty dt.$$

Notice here that it is crucial to produce estimates for $\nabla\phi$.

**Wasserstein-1 ($\mathbf{d}_1$) estimates.** To prove (10), we choose $\mathcal{X} = \mathcal{Y} = \{\psi : \Omega \to \mathbb{R}, \|\nabla\psi\|_\infty \le 1\}$, the space of 1-Lipschitz functions. We obtain (22) again, except the terminal data $\psi$ also has Lipschitz constant equal to 1. Applying Cauchy-Schwarz in space and time implies the second term is bounded

$$\int_0^T \int_\Omega m^2 \nabla\phi \cdot (b^2 - b^1)dxdt \le \left(\int_0^T \int_\Omega |\nabla\phi|^2 m^2 dxdt\right)^{\frac{1}{2}} \|b^1 - b^2\|_{L^2(m^2)} \qquad (25)$$
$$\le T \sup_{0 \le t \le T} \|\nabla\phi(t, \cdot)\|_\infty \|b^1 - b^2\|_{L^2(m^2)}.$$

We again see that we require estimates for $\nabla\phi$, and we need them to be bounded.

### 4.3 Bernstein estimates from HJB theory provide gradient estimates

To obtain estimates for $\nabla\phi$, we could differentiate (18) to derive a PDE for $\nabla\phi$. The resulting PDE, however, will have $\partial_t(\nabla\phi)$ grow linearly with $\nabla\phi$, and so if we apply (reverse) Gronwall's inequality, the resulting estimates for $\nabla\phi$ will grow exponentially in time. To avoid this exponential time dependence, we first perform a Hopf-Cole transform $u = -2\log\phi$ on (18) to derive the HJB equation for $u$ [9]

$$-\partial_t u - \Delta u + \frac{1}{2}|\nabla u|^2 + b \cdot \nabla u = 0, \quad u(T, x) = -2\log(\psi(x)). \qquad (26)$$

Here we provide an example of classical Bernstein estimates (Proposition D.1 and Corollary D.2) to obtain bounds for $\nabla\phi$ without using Gronwall's inequality [10]. The main idea is to derive a PDE (60) for the function $z = \frac{1}{2}|\nabla u|^2$ by taking the gradient of (26) and then taking the inner product with $\nabla u$. Then by showing that the function $w(t, x) = z - Cu$ for sufficiently large $C$ attains its maximum at $(T, x_0)$, we can show that

$$z(t, x) \le w(T, x_0) + Cu(t, x) \le \frac{1}{2}|2\nabla\log\psi|^2 + C\|2\log\psi\|_\infty + C\|u\|_\infty. \qquad (27)$$

Using the maximum principle for (18), we find $\|u\|_\infty = \|2\log\psi\|_\infty$, yielding $z(t, x) \le C\|\log\psi\|_{C^1}$. Assuming that $\psi$ is Lipschitz continuous implies boundedness of $z$ and therefore $\nabla\phi$ for all time. A similar result holds when $\psi \in L^\infty$ only (see (59)). Detailed proofs and related boundes are provided in Proposition D.1. Applications of the bounds to derive the WUP is provided in Section A.1.

**Remark 4.1** (The regularizing role of stochasticity). *In our analysis, the stochasticity in SGMs provides two types of regularizing effects. The first is early stopping, which adds a small amount of Gaussian noise of the data distribution. This, in effect, is equivalent to running the noising process*

*for a short amount of time and immediately mollifies the initial empirical distribution so that it has a smooth density. Second, the Laplacian is the key mechanism that regularizes the test function in* (18) *[9], which then allows us to bound, for example, the stronger $\| \cdot \|_{L^1}$ norm by the weaker $\mathbf{d}_1$-norm. Recall that in PDE theory, the stochasticity of a generative flow manifests as the Laplacian operator in the Fokker-Planck and Kolmogorov backward equations [9]. Without the Laplacian the regularizing the test functions in* (18) *would not be possible in general and, in fact, we would not have access to long time behavior results.*

# 5    Proof sketches — Score-based generative models are robust to errors

We now provide sketches of the proofs for the main generalization bounds in Theorems 3.2 and 3.3. The full proofs are provided in Sections A.2 and A.3, respectively.

## 5.1    Theorem 3.2: ESM generalization bound

Theorem 3.2 is a generalization bound with respect to error from the score function approximation with respect to the ESM objective ($e_3$) and the choice of reference measure ($e_4$). Assuming we have an (uniform-in-time) $L^2$-close approximation of the score function, first apply the WUP Theorem 3.1 with $m_1 = \frac{1}{\text{vol}(R\mathbb{T}^d)}$ and $m_2 = \eta^\pi(T, \cdot)$. The distance between $m_1$ and $m_2$ can be expressed in terms of $\pi$ by studying the long time behavior of periodic solutions to the heat equation. Proposition D.3 shows the heat equation is a contraction under $\mathbf{d}_1$. Applying it to $m^1$ and $m^2$ yields the desired result.

While Theorem 3.2 has no explicit assumptions on $\pi$, the assumption that the approximate score function $\mathbf{s}_\theta$ is close in $L^2$ to the true score implicitly implies regularity of $\pi$. Specifically, if $\mathbf{s}_\theta$ is Lipschitz (which is true for neural networks in practice) and satisfies $\mathcal{J}(\pi, \theta) < e_{nn}$, then $\pi$ must have finite entropy. This implies that $\pi$ necessarily has a density. See Proposition B.8 for the formal statement and proof.

## 5.2    Theorem 3.3: Pointwise DSM generalization bound

In practice, the score function is learned via a Monte Carlo approximation of the DSM objective (2). To avoid overfitting to the kernel estimator and memorizing the dataset [25], the time integral is taken only over $s \in [\epsilon, T]$ where $\epsilon$ is the early stopping parameter. This is equivalent to training with the ESM objective with the true score replaced with the kernel approximation at time $\epsilon$ [20, 27]. To derive generalization bounds of SGMs trained via DSM, we establish the relationships between (1) the mollified distribution $\pi^\epsilon = \Gamma(\epsilon) \star \pi$ and the true distribution $\pi$, and (2) the DSM objective $J_{DSM}(\eta^{N,\epsilon}, \theta) = \mathcal{J}(\eta^{N,\epsilon}, \theta)$ and the ESM objective with respect to the score function of the mollified distribution $\mathcal{J}(\eta^{\pi^\epsilon}, \theta)$. Formally, this strategy involves bounding the following two terms

$$\underbrace{\mathbf{d}_1(\pi, m_g(T))}_{\text{Generalization bound w.r.t true distribution}} \leq \underbrace{\mathbf{d}_1(\pi, \pi^\epsilon)}_{\text{Early stopping error } (e_5)} + \underbrace{\mathbf{d}_1(\pi^\epsilon, m_g(T))}_{\text{Generalization bound w.r.t. mollified distribution}}. \tag{28}$$

For the early stopping error, we use the regularizing properties of the heat equation to show that $\mathbf{d}_1(\pi, \pi^\epsilon) \leq C\sqrt{\epsilon}$, where $C$ only depends on the dimension $d$. This implies that, measured in $\mathbf{d}_1$, early stopping only incurs a nominal $C\sqrt{\epsilon}$ error *even if the $\pi$ does not admit a density*. This result would not possible if we were to study generalization error in terms of KL or TV directly.

A bound for the second term is obtained by comparing the ESM objective value between the learned score function and the true score function of the early stopped distribution, $\mathcal{J}(\eta^{\pi^\epsilon}, \theta)$ to the DSM objective $\mathcal{J}(\eta^{N,\epsilon}, \theta)$. We present Theorem B.5 and its corollary, which under the assumption that $\eta^{N,\epsilon}$ and $\eta^{\pi^\epsilon}$ have a lower bound $\delta$, state that if $\mathcal{J}(\eta^{N,\epsilon}, \mathbf{s}_\theta) < e_{nn}$, then $\mathcal{J}(\eta^{\pi^\epsilon}, \mathbf{s}_\theta) < e'_{nn}$ with

$$e'_{nn} = e_{nn} + C\left(1 + \frac{|\log(\delta)|}{\sqrt{\epsilon}} + \frac{1}{\sqrt{T}} + T\|\mathbf{s}_\theta\|^2_{C^2([0,T]\times\Omega)}\right)\mathbf{d}_1(\pi^N, \pi). \tag{29}$$

The main idea behind Theorem B.5 is to note that the difference between the ESM and DSM objective functions can be written as a difference in ISM objective functions plus the entropy difference between $\eta^{N,\epsilon}$ and $\eta^{\pi^\epsilon}$. Details for bounding this term is provided in Proposition B.2 and Lemma B.3.

To arrive at the final result, apply the WUP theorem to derive generalization bounds for $\mathbf{d}_1(\pi^\epsilon, m_g(T))$ under the assumption that $\mathcal{J}(\eta^{\pi^\epsilon}, \theta) < e'_{nn}$, along with (29). Finally, combine this result with the error due to the early stopping $\mathbf{d}_1(\pi, \pi^\epsilon)$. Full details of this proof is provided in Section A.3.

# 6 Discussion: PDE regularity theory and UQ for generative modeling

Our main contribution is the study of generalization in score-based generative models from the perspective of uncertainty quantification. The regularity theory of nonlinear PDEs is the key technical tool that produces our results. We emphasize that the tools we use here can be used generative models beyond SGMs, and that we have not pushed our analysis to the limits of our tools in SGMs. Moreover, we also emphasize some downstream UQ applications for SGMs that may be of future interest.

## 6.1 The significance of the *regularizing* properties of SGMs

A surprising result of our work is deriving bounds for the stronger $L^1$ distance in terms of the weaker $\mathbf{d}_1$ distance (see (22) and Theorem 3.2). The key insight (Equation (21)) is that the evolution of observables defined by the KBE (18) *regularizes* the test function. We have not fully exploited the regularizing effects of (18) in this paper, as we only focused on $L^1$ and $\mathbf{d}_1$ estimates.

**Improved bounds in Sobolev spaces $H^s$.** To illustrate other extensions and choices of $\mathcal{Y}$ in (21), observe that in the trivial case when $b^1 = b^2 = 0$, (19) simplifies to $\int \lambda(T,x)\psi(x)dx = \int \lambda(0,x)(\Gamma(T) \star \psi)(x)dx$, where $\Gamma$ is the heat kernel. If $\|\psi\|_\infty \leq 1$, then $\Gamma \star \psi(T) \in C^\infty$ and we have estimates of the form $\mathbf{d}_1(m^1(T), m^2(T)) \leq C(s,T,d)\|m_1 - m_2\|_{H^{-s}}$ for all $s \in \mathbb{N}$. When $b^1, b^2$ are not identically zero we still expect such estimates, though they will depend on the regularity of $b^1$. To highlight the importance of regularizing effects, note that by [31], if $\pi \in \mathcal{P}(\mathbb{T}^d)$, then for the empirical measure $\pi^N$, we expect $\mathbf{d}_1(\pi, \pi^N) \lesssim \frac{1}{N^{\frac{1}{d}}}$. However if $s > \frac{d}{2}$, $\|\pi - \pi^N\|_{H^{-s}} \lesssim \frac{1}{\sqrt{N}}$. This suggests that improved regularity may influence overcoming the curse of dimensionality.

## 6.2 A connection to likelihood-free inference

Computing expectations with respect to posterior distributions is a key task in Bayesian inference. Generative modeling, in particular, has a key role in future developments of *likelihood-free* inference [32, 33, 34, 35]. For generative models to be trustworthy for inference, they must to be shown to be robust. The WUP theorem provides error bounds for approximating expectations with respect to some true unknown distribution, and may be significant for SGMs in likelihood-free inference.

For example, suppose we wish to estimate $\mathbb{E}_\pi h$ for some distribution $\pi$ and observable $h$, and an SGM $m_g(T)$ is constructed to approximate the expectation. Bounds of the form (6), such as the WUP theorem, translate into guarantees on the expectations:

$$\left|\mathbb{E}_\pi h - \mathbb{E}_{m_g(T)} h\right| \leq \mathbf{d}(m_g(T), \pi) \leq \mathcal{F}(e_1, e_2, e_3, e_4, e_5). \tag{30}$$

In the context of SGM, the inequalities are *a posteriori* bounds, meaning they can be computed after learning the model. Additional regularity on $h$ may yield improved guarantees.

## 6.3 Enabling distributionally robust optimization (DRO)

Robust UQ methods are based on the perspective that learning any complex model will typically involve multiple sources of uncertainty due to modeling choices, model reduction or learning from imperfect data. These uncertainties are not just in parameters but are inherently present in the mathematical model itself and will propagate to any predictions. There is substantial related work in recent years using a distributional robustness perspective, [12], using divergences or probability metrics and their variational representations to quantify the impact of model uncertainty around a baseline model that may be either learned (e.g. a generative model) or could be just an empirical distribution from an unknown true distribution. The approach can generally be described as quantifying an uncertainty set around the baseline model that the worst-case distribution belongs to via some neighborhood defined in terms of a probability divergence [13, 14], a Wasserstein distance [15] or maximum mean discrepancy [16]. There are, however, drawbacks to each of these approaches: a divergence ball contains only distributions with the same support as the baseline, while the uncertainty set may be hard to determine practically. The WUP Theorem is a related robust UQ notion where the uncertainty ball is in an IPM, e.g. 1-Wasserstein, MMD, or TV. However, it allows us to bypass the robust UQ for stochastic processes relying on restrictive, path-space probability divergence-based approaches or Girsanov's Theorem, [36, 37]. Furthermore, the WUP Theorem bounds use PDE theory to provide a computable uncertainty set, as we also demonstrate in the case of DSM, see Theorem 3.3.

## Acknowledgements

M.K. and B.Z. are partially funded by AFOSR grant FA9550-21-1-0354. M.K. is partially funded by NSF DMS-2307115 and NSF TRIPODS CISE-1934846.

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

# A  Proofs of main results

Throughout the proofs we will make frequent use of the fact that if $\Gamma : [0, \infty) \times \Omega \to \mathbb{R}$ denotes the heat kernel on $\Omega = \mathbb{R}^d$ or $\Omega = R\mathbb{T}^d$, then there exists a dimensional constant $C = C(d) > 0$ such that

$$|\nabla\Gamma(t) \star g| \le C \frac{\|g\|_\infty}{\sqrt{t}}. \tag{31}$$

Furthermore, recall that constants are subject to change from line to line but always maintain a variable dependence as in the corresponding statement.

## A.1  Theorem 3.1

The proof of Theorem 3.1 follows the strategy outlined in Section 4 and uses the different gradient estimates from the HJB equations provided in Section D.1.

*Proof.* Let $\lambda = m^1 - m^2$, which satisfies

$$\begin{cases} \partial_t \lambda - \Delta\lambda - \operatorname{div}(\lambda b^1 + m^2(b^2 - b^1)) = 0 \text{ in } (0, T) \times R\mathbb{T}^d, \\ \lambda(0) = m_1 - m_2 \text{ in } R\mathbb{T}^d. \end{cases} \tag{32}$$

For a fixed $\psi : \Omega \to \mathbb{R}$, for which we will specify bounds latter, let $\phi : [0, T] \times \Omega \to \mathbb{R}$ be given by

$$\begin{cases} -\partial_t \phi - \Delta\phi + b^1 \cdot \nabla\phi = 0 \text{ in } [0, T) \times \Omega, \\ \phi(T, x) = \psi(x) \text{ in } \Omega. \end{cases} \tag{33}$$

Testing against $\phi$ in (32) and using (33), we obtain

$$\int \lambda(T, x)\psi(x)dx = \int \lambda(0)\phi(0, x)dx - \int_0^T \int m^2(s)\nabla\phi(s) \cdot (b^2 - b^1)(s)dxds. \tag{34}$$

- Proof of (8) and (9): Assume that $\|\psi\|_\infty \le 1$ which then implies

$$-1 \le \phi \le 1.$$

Since $\int d\lambda(t) = 0$ for all $t \in [0, T]$ up to adding a constant we can assume without loss of generality in (34), that

$$1 \le \psi \le 3 \text{ and } 1 \le \phi \le 3.$$

Then from estimate (63) in Corollary D.2, we have

$$(T - t)\|\nabla\phi(t)\|_\infty^2 \le C(T\|\nabla b\|_\infty + 1)$$

$$\implies \|\nabla\phi(t)\|_\infty \le C \frac{\sqrt{T\|\nabla b^1\|_\infty + 1}}{\sqrt{T - t}}.$$

Next, we note that

$$\int \lambda(T, x)\psi(x)dx = \int \lambda(0)\phi(0, x)dx - \int_0^T \int m^2(s)\nabla\phi(s) \cdot (b^2 - b^1)(s)dxds$$

$$\le \mathbf{d}_1(m^1, m^2)\|\nabla\phi(0)\|_\infty + \int_0^T \left(\int |\nabla\phi(s)|^2 m^2(s)dx\right)^{\frac{1}{2}} \left(\int m^2(s)|b^2 - b^1|^2(s)dx\right)^{\frac{1}{2}} ds$$

$$\le \mathbf{d}_1(m^1, m^2)\|\nabla\phi(0)\|_\infty + \sup_{0 \le t \le T} \|(b^2 - b^1)(t)\|_{L^2(m^2(t))} \int_0^T \|\nabla\phi(s)\|_\infty ds,$$

$$\le C(\sqrt{T\|\nabla b^1\|_\infty} + 1)\left(\frac{\mathbf{d}_1(m^1, m^2)}{\sqrt{T}} + \sqrt{T} \sup_{0 \le t \le T} \|(b^2 - b^1)(t)\|_{L^2(m^2(t))}\right).$$

Taking the supremum over $\|\psi\|_\infty \le 1$ yields

$$[\|m^2(T) - m^1(T)\|_{L^1(\Omega)} \le C(\sqrt{T\|\nabla b^1\|_\infty} + 1) \times$$

$$\left(\frac{\mathbf{d}_1(m^1, m^2)}{\sqrt{T}} + \sqrt{T} \sup_{0 \le t \le T} \|(b^2 - b^1)(t)\|_{L^2(m^2(t))}\right).$$

Finally, if instead we used the bound

$$\int \lambda(0)\phi(0,x)dx \le \|\lambda(0)\|_{L^1(\Omega)}\|\phi\|_\infty \le C\|m_1 - m_2\|_{L^1(\Omega)}$$

we obtain (9).

- Proof of (10): Let $\Omega = R\mathbb{T}^d$ and consider a $\psi$ with $\|\nabla\psi\|_\infty \le 1$. Without loss of generality we can assume $\psi(0) = R + 1$ and thus

$$\psi(x) = \psi(x) - \psi(0) + \psi(0) \ge -|x - 0| + R + 1 \ge -R + R + 1 = 1$$

and similarly

$$\psi(x) \le 2R + 1.$$

Therefore by maximum principle we have that

$$1 \le \phi(t,x) \le 2R + 1,$$

and so

$$\|\psi\|_{C^1} = \|\psi\|_\infty + \|\nabla\psi\|_\infty \le C(1 + R).$$

From (34) we have

$$\int \lambda(T,x)\psi(x)dx \le \mathbf{d}_1(m_1,m_2)\|\nabla\phi(0)\|_\infty + \sup_{0 \le t \le T}\|\nabla\phi(t)\|_\infty\|b^1 - b^1\|_{L^2(m^2)}.$$

Thus by Corollary D.2, estimate (62) we obtain

$$\int \lambda(T,x)\psi(x)dx \le C\sqrt{(1 + \|\nabla b\|_\infty)\|v\|_{C^1}^3}(\mathbf{d}_1(m_1,m_2) + \|b^1 - b^1\|_{L^2(m^2)})$$

$$\le CR^{\frac{3}{2}}(1 + \sqrt{\|\nabla b^1\|_\infty})\Big(\mathbf{d}_1(m_1,m_2) + \|b^1 - b^1\|_{L^2(m^2)}\Big),$$

which after taking the supremum over $\|\nabla\psi\|_\infty \le 1$ yields (10).

$\square$

## A.2 Theorem 3.2

The proof of Theorem 3.2 follows by an application of Theorem 3.1.

*Proof.* Let $b^1 = \mathbf{s}_\theta, m_1 = \frac{1}{\text{vol}(R\mathbb{T}^d)}$ and $b^2 = \nabla\log(\eta^\pi), m_2 = \eta^\pi(T)$. Note that if $m^i, i = 1, 2$ solve

$$\begin{cases} \partial_t m^i - \Delta m^i - \text{div}(m^i b^i) = 0 \text{ in } [0,T] \times R\mathbb{T}^d, \\ m^i(0) = m_i \text{ in } [0,T] \times R\mathbb{T}^d, \end{cases} \tag{35}$$

then $m^1 = m_g, m^2(t) = \eta^\pi(T - t)$. Thus, by an application of Theorem 3.1 we obtain

$$\mathbf{d}_1(m_g(T),\pi) \le CR^{\frac{3}{2}}(1 + \sqrt{\|\nabla\mathbf{s}_\theta\|_\infty})(\mathbf{d}_1(\frac{1}{\text{vol}(R\mathbb{T}^d)},\eta^\pi) + \|\mathbf{s}_\theta - \nabla\log(\eta^\pi)\|_{L^2(\eta^\pi)})$$

$$\le CR^{\frac{3}{2}}(1 + \sqrt{\|\nabla\mathbf{s}_\theta\|_\infty})\Big((Re^{-\frac{\omega T}{R^2}}\mathbf{d}_1\Big(\pi,\frac{1}{\text{vol}(R\mathbb{T}^d)}\Big) + \sqrt{e_{nn}}\Big),$$

where in the last inequality we used Proposition D.3. The second claim follows again by Theorem 3.1 and the fact that

$$\mathbf{d}_1(\nu_1,\nu_2) = \sup_{\|\nabla g\|_\infty \le 1}\int_{R\mathbb{T}^d} gd(\nu_1 - \nu_1) \le R\|\nu_2 - \nu_1\|_{L^1}.$$

$\square$

## A.3 Theorem 3.3

The proof of Theorem 3.3 follows the same strategy as in Theorems 3.1 and 3.2. The main technical difference is that we only have a bound between our generated score function $\mathbf{s}_\theta$ and the score function generated by the sample $\nabla \log(\eta^{N,\epsilon})$ of the form

$$\int_0^T \int |\mathbf{s}_\theta - \nabla \log(\eta^{N,\epsilon})|^2 d\eta^{N,\epsilon}(s)ds \le e_{nn}$$

and so we need to produce a bound between $\mathbf{s}_\theta$ and the true score function $\nabla \log(\eta^\epsilon)$. This last step is shown in Section B and in particular Theorem B.5.

*Proof of Theorem 3.3.* Consider the function $\eta^{\pi,\epsilon} : [0,T] \times R\mathbb{T}^d \to \mathbb{R}$ given by

$$\begin{cases} \partial_t \eta^{\pi,\epsilon} - \Delta \eta^{\pi,\epsilon} = 0 \text{ in } (0,T) \times R\mathbb{T}^d, \\ \eta^{\pi,\epsilon}(0) = \pi^\epsilon \text{ in } R\mathbb{T}^d. \end{cases} \tag{36}$$

Define the drifts

$$\mathbf{b}_{\theta^*}(t,x) := \mathbf{s}_{\theta^*}(T-t,x)$$

and

$$\mathbf{b}^{\pi^\epsilon}(t,x) := \nabla \log(\eta^{\pi^\epsilon})(T-t,x),$$

and let $m^\epsilon(t,x) = \eta^{\pi,\epsilon}(T-t,x)$ which satisfies

$$\begin{cases} \partial_t m^\epsilon - \Delta m^\epsilon - \operatorname{div}(m^\epsilon \mathbf{b}^{\pi^\epsilon}(t,x)) = 0, \\ m^\epsilon = \eta^{\pi,\epsilon}(T,x). \end{cases} \tag{37}$$

Finally, we consider the distribution of our generated sample $m_g : [0,T] \times \Omega \to \mathbb{R}$ which is given by

$$\begin{cases} \partial_t m_g - \Delta m_g - \operatorname{div}(m_g \mathbf{b}_{\theta^*}) = 0 \text{ in } (0,T] \times \Omega, \\ m_g(0) = \frac{1}{\operatorname{vol}(R\mathbb{T}^d)} \text{ in } \Omega. \end{cases} \tag{38}$$

We have the following

$$\mathbf{d}_1(\pi, m_g(T)) \le \boxed{\mathbf{d}_1(\pi, \pi^\epsilon)}_1 + \boxed{\mathbf{d}_1(\pi^\epsilon, m_g(T))}_2, \tag{39}$$

We will bound each of the terms separately.

1. We recall that $\pi^\epsilon = \Gamma(\epsilon) \star \pi$ where $\Gamma$ is the heat kernel. Therefore, by the dual formulation of $\mathbf{d}_1$ we have that

$$\mathbf{d}_1(\pi, \pi^\epsilon) = \sup_{\|\nabla g\|_\infty \le 1} \int g(x)d(\pi^\epsilon - \pi) = \sup_{\|\nabla g\|_\infty \le 1} \int (\Gamma(\epsilon) \star g(x) - g(x))d\pi(x).$$

Although estimates on $(\Gamma(\epsilon) \star g(x) - g(x))$ for a function $\|\nabla g\|_\infty \le 1$ are classical for the readers convenience we provide a quick proof. Let $v(t) = \Gamma(t) \star g$, which solves

$$\begin{cases} \partial_t v - \Delta v = 0, \\ v(0) = g. \end{cases} \tag{40}$$

Then for all $x \in \Omega$,

$$|v(\epsilon,x) - v(0,x)| \le \int_0^\epsilon |\partial_t v(s,x)|ds = \int_0^\epsilon |\Delta v(s,x)|ds$$

$$= \int_0^\epsilon |(\nabla \Gamma(s) \otimes \star \nabla g)(x)|ds \le C\|\nabla g\|_\infty \int_0^\epsilon \frac{1}{\sqrt{s}}ds \le C\sqrt{\epsilon}.$$

Where in the above we used estimate (31). Thus, we have

$$\mathbf{d}_1(\pi, \pi^\epsilon) \le C\sqrt{\epsilon},$$

where $C = C(d) > 0$.

2. First we note that $\pi^\epsilon = m^\epsilon(T)$. Thus applying Theorem 3.1 for

$$m^1 = m_g, b^1 = \mathbf{b}_{\theta^*}$$

$$m^2 = m^\epsilon, b^2 = \mathbf{b}^{\pi^\epsilon}$$

yields

$$\mathbf{d}_1(m_g(T), \pi^\epsilon) = \mathbf{d}_1(m_g(T), m^\epsilon(T))$$

$$\lesssim R^{\frac{3}{2}}(1 + \sqrt{\|\nabla b^1\|_\infty})(\mathbf{d}_1(m^\epsilon(0), \frac{1}{\text{vol}(R\mathbb{T}^d)}) + \sqrt{T}\|\mathbf{b}^{\pi^\epsilon} - \mathbf{b}_{\theta^*}\|_{L^2(m^\epsilon)}).$$

By Proposition D.3, we have

$$\mathbf{d}_1(m^\epsilon(0), \frac{1}{\text{vol}(R\mathbb{T}^d)}) = \mathbf{d}_1(\eta^{\pi,\epsilon}(T), \frac{1}{\text{vol}(R\mathbb{T}^d)}) \le CRe^{-\frac{\omega t}{R^2}}\mathbf{d}_1(\pi^\epsilon, \frac{1}{\text{vol}(R\mathbb{T}^d)})$$

$$\le CR^2e^{-\frac{\omega t}{R^2}}.$$

Finally, from Theorem B.5

$$\|\mathbf{b}^{\pi^\epsilon} - \mathbf{b}_{\theta^*}\|_{L^2(m^\epsilon)}^2 = \mathcal{J}(\eta^{\pi^\epsilon}, \theta^*) \le e'_{nn}$$

$$= e_{nn} + C\left(1 + \frac{|\log(\delta)|}{\sqrt{\epsilon}} + \frac{1}{\sqrt{T}} + T\|\mathbf{s}_\theta\|_{C^2([0,T]\times\Omega)}^2\right)\mathbf{d}_1(\pi^N, \pi)$$

Therefore, if $T \ge 1$, putting everything together, we have shown that up to a dimensional constant

$$\mathbf{d}_1(\pi, m_g(T)) \le C\left(\sqrt{\epsilon} + R^{\frac{3}{2}}\left(1 + \sqrt{\|\nabla b^1\|_\infty}\right) \times\right.$$

$$\left.\left[R^2e^{-\frac{\omega T}{R^2}} + \sqrt{T\left(e_{nn} + C\left(1 + \frac{|\log(\delta)|}{\sqrt{\epsilon}} + T\|\mathbf{s}_\theta\|_{C^2([0,T]\times\Omega)}^2\right)\mathbf{d}_1(\pi^N, \pi)\right)}\right]\right)$$

$\square$

# B    Analysis of Score Matching functionals

In this section, we gather all the technical results regarding the connections between ISM, DSM, and ESM score matching functionals. For the readers convenience we recall some of our notations. Let $\Omega \subset \mathbb{R}^d$ and $m_0 \in \mathcal{P}(\Omega)$. We will denote by $\rho^{m_0} : [0, T] \times \Omega \to [0, \infty)$ the solution of

$$\begin{cases} \partial_t \rho^{m_0} - \Delta\rho^{m_0} = 0 \text{ in } (0, T] \times \Omega, \\ \rho^{m_0}(0) = m_0 \text{ in } \Omega, \end{cases} \tag{41}$$

and we may drop the superscipt $m_0$ when it is clear from context.

First we state a simple result that justifies the formula

$$\mathcal{J}(\rho^{m_0}, \theta) = \mathcal{J}_L(\rho^{m_0}, \theta) + 4\|\nabla\sqrt{\rho^{m_0}}\|_2^2, \tag{42}$$

which formally follows by

$$\mathcal{J}(\rho, \theta) = \int_0^T \int \left(|\mathbf{s}_\theta|^2 d\rho(s) - 2\mathbf{s}_\theta \cdot \nabla\log(\rho) + |\nabla\log(\rho)|^2\right)d\rho(s)\mathrm{d}s$$

$$= \mathcal{J}_L(\rho, \theta) + \int_0^T \int \frac{|\nabla\rho|^2}{\rho}dxds$$

$$= \mathcal{J}_L(\rho) + 4\|\nabla\sqrt{\rho}\|_2^2.$$

However $\|\nabla\sqrt{\rho^{m_0}}\|_2$ may not be finite, thus we record the exact statement along with some technical facts for later use in the following Proposition.

**Proposition B.1.** *Let $m_0$ be a probability density in $\Omega$ such that $m_0(x)\log(m_0(x)) \in L^1(\Omega)$ and $\rho : [0,T] \times \Omega \to \mathbb{R}$ be given by* (41). *Then, the following hold:*

1. *There exists a universal constant $C > 0$, such that*

$$\sup_{t \in [0,T]} \|\rho(t)\log(\rho(t))\|_{L^1(\Omega)} + \|\nabla\sqrt{\rho}\|^2_{L^2([0,T]\times\Omega)} \leq CT\|m_0\log(m_0)\|_{L^1(\Omega)},$$

   *and thus* (42) *holds.*

2. $4\|\nabla\sqrt{\rho}\|^2_2 = \int_\Omega m_0\log(m_0) - \rho(T)\log(\rho(T))dx.$

*Proof.* Testing (41) against $f'(\rho)$ for a smooth function $f$, we obtain

$$\int f(\rho(T,x))dx + \int_0^T \int f''(\rho)|\nabla\rho|^2 dxdt = \int f(m_0(x))dx$$

and the choice of $f(x) = x\log(x)$ yields

$$\int \rho(T,x)\log(\rho(T,x))dx + \int_0^T \int \frac{|\nabla\rho|^2}{\rho} dxdt = \int m_0(x)\log(m_0(x))dx,$$

which shows the second claim. Finally, if $m_0\log(m_0) \in L^1$ this yields estimates on $\sup_{t \in [0,T]} \|\rho(t)\log(\rho(t))\|_1, \|\nabla\sqrt{\rho}\|^2_2$ since

$$\int |\rho(T,x)\log(\rho(T,x))|dx = \int \rho(T,x)\log(\rho(T,x))dx + 2\int (\rho(T,x)\log(\rho(T,x)))^- dx$$

$$\leq \int \rho(T,x)\log(\rho(T,x))dx + 2e^{-1}|\Omega|,$$

where we used the fact that $x\log(x) \geq -\frac{1}{e}$. $\qquad\qquad\square$

The rest of the subsection justifies the steps outlined in Subsection 5.2. Namely starting from a estimate of the form

$$\mathcal{J}(\eta^{N,\epsilon}, \theta) \leq e_{nn}$$

our goal is to obtain an estimate of the form

$$\mathcal{J}(\eta^\epsilon, \theta) \leq e'_{nn}.$$

For this we use formula (42). In particular we first use the linear dependence of $\mathcal{J}_L$ with respect to the underlying measure to bound

$$|\mathcal{J}_L(\eta^{N,\epsilon}, \theta) - \mathcal{J}_L(\eta^\epsilon)|.$$

Then, instead of comparing the gradients on the difference

$$\|\nabla\log(\eta^{N,\epsilon})\|^2_2 - \|\nabla\log(\eta^\epsilon)\|^2_2$$

we use Proposition B.1 from which it is enough to bound the entropy terms. The next two Propositions provide these results along with some other technical facts.

**Proposition B.2.** *Let $T > 0$ and $\pi^i$ for $i = 1,2$ denote two probability measures in $\Omega$ such that $\|\pi^i\log(\pi^i)\|_{L^1} < \infty$ and $\rho^i$ the corresponding solutions to* (41). *Then, there exists a dimensional constant $C > 0$ such that*

$$|\mathcal{J}_L(\rho^2, \theta) - \mathcal{J}_L(\rho^1, \theta)| \leq CT \sup_{t \in [0,T]} \mathbf{d}_1(\rho^2(t), \rho^1(t))\|\mathbf{s}_\theta\|^2_{C^2([0,T]\times\Omega)}$$

$$\leq CT\mathbf{d}(\pi^1, \pi^2)\|\mathbf{s}_\theta\|^2_{C^2([0,T]\times\Omega)}.$$

*In the above $\mathbf{s}_\theta$ indicates the score function approximation.*

*Proof.* We recall that

$$\mathcal{J}_L(\rho^i, \theta) = \int_0^T \mathbb{E}_{\rho^i}\left[\frac{1}{2}|\mathbf{s}_\theta|^2 + \operatorname{div}(\mathbf{s}_\theta)\right]ds = \int_0^T \int \frac{1}{2}|\mathbf{s}_\theta|^2 + \operatorname{div}(\mathbf{s}_\theta)d\rho^i(s)ds.$$

Therefore, if

$$g_\theta = \frac{1}{2}|\mathbf{s}_\theta|^2 + \operatorname{div}(\mathbf{s}_\theta)$$

$$|\mathcal{J}_L(\rho^2, \mathbf{s}_\theta) - \mathcal{J}_L(\rho^1, \mathbf{s}_\theta)| = \left|\int_0^T \int \frac{1}{2}|\mathbf{s}_\theta|^2 + \operatorname{div}(\mathbf{s}_\theta)d(\rho^2(s) - \rho^1(s))ds\right|$$

$$= \left|\int_0^T \int g_\theta d(\rho^2 - \rho^1)(s)ds\right| \leq \sup_{0 \leq t \leq T} \|\nabla g_\theta\|_\infty \int_0^T \mathbf{d}_1(\rho^2(s), \rho^1(s))ds$$

$$\leq T \sup_{0 \leq t \leq T} \|\nabla g_\theta\|_\infty \sup_{0 \leq t \leq T} \mathbf{d}_1(\rho^2(t), \rho^1(t)).$$

Since

$$\|\nabla g_\theta\|_\infty \leq \|\mathbf{s}_\theta\|_{C^2}^2$$

to conclude we need to show that

$$\sup_{0 \leq t \leq T} \mathbf{d}_1(\rho^2(t), \rho^1(t)) \leq \mathbf{d}_1(\pi^2, \pi^1).$$

However this follows from our Lemma 3.1 applied for $b^1 = b^2 = \vec{0}$ and $m^i = \pi^i$ for $i = 1, 2$. $\quad\square$

**Lemma B.3.** *Let $\pi^\epsilon, \hat{\pi}^{N,\epsilon}, 0 < \delta < \epsilon$ be as in Theorem 3.3 with $\epsilon < 1$. There exists a dimensional constant $C = C(d) > 0$ such that*

$$\mathbf{d}_1(\hat{\pi}^{N,\epsilon}, \pi^\epsilon) \leq \mathbf{d}_1(\pi^N, \pi) \tag{43}$$

$$\|\pi^\epsilon - \hat{\pi}^{N,\epsilon}\|_{L^1(\Omega)} \leq C\frac{\mathbf{d}_1(\pi^N, \pi)}{\sqrt{\epsilon}}, \tag{44}$$

*and*

$$\|\log(\pi^\epsilon)\pi^\epsilon - \log(\hat{\pi}^{N,\epsilon})\hat{\pi}^{N,\epsilon}\|_{L^1(\Omega)} \leq C(1 + |\log(\delta)|)\frac{\mathbf{d}_1(\pi^N, \pi)}{\sqrt{\epsilon}}. \tag{45}$$

*Moreover, let $C > 0$ denote the constant from Proposition D.3 and $T \geq R^2$ large enough such that*

$$2CR^{-d}e^{-\frac{\omega T}{R^2}} \leq \frac{1}{2}\frac{1}{vol(R\mathbb{T}^d)},$$

*then up to a dimensional constant $C > 0$ we have*

$$\int \log(\eta^{N,\epsilon}(T))\eta^{N,\epsilon}(T) - \log\eta(T))\eta(T)dx \leq \frac{C}{\sqrt{T}}\left(1 + d\log(R)\right)\mathbf{d}_1(\pi, \hat{\pi}^N). \tag{46}$$

*Proof.* Let $\rho^{N,\epsilon}, \rho^\epsilon : [0, T] \times \Omega \to \mathbb{R}$, be solutions to

$$\begin{cases} \partial_t \rho - \Delta\rho = 0 \text{ in } \Omega \times (0, \epsilon), \\ \rho(0) = \rho_0, \end{cases} \tag{47}$$

for $\rho_0 = \pi, \rho_0 = \pi^N$ respectively. Since $\pi^\epsilon = \Gamma(\epsilon) \star \pi, \hat{\pi}^{N,\epsilon} = \Gamma(\epsilon) \star \pi^N$ we note that $\pi^\epsilon = \rho^\epsilon(\epsilon, x)$ and $\hat{\pi}^{N,\epsilon} = \rho^{\epsilon,N}(\epsilon, x)$. Using the same adjoint method as in the proof of Theorem 3.1 for a function $\psi : \Omega \to \mathbb{R}$ we consider the function $\phi : [0, \epsilon] \times \Omega \to \mathbb{R}$ given by

$$\begin{cases} -\partial_t \phi - \Delta\phi = 0 \\ \phi(\epsilon) = \psi. \end{cases} \tag{48}$$

Testing against $\phi$ in the equation for $\lambda = \rho^\epsilon - \rho^{\epsilon,N}$ yields

$$\int(\hat{\pi}^{N,\epsilon} - \pi^\epsilon)\psi(x)dx = \int \phi(0, x)d(\pi - \pi^N)(x).$$

First we consider $\|\psi\|_\infty \leq 1$ which from (31), yields

$$\|\nabla\phi(0)\|_\infty \leq \frac{\|\psi\|_\infty}{\sqrt{\epsilon}} \leq \frac{C}{\sqrt{\epsilon}}.$$

Thus, for any $\|\psi\|_\infty \leq 1$ we obtain

$$\int (\hat{\pi}^{N,\epsilon} - \pi^\epsilon)\psi(x)dx \leq \frac{C\mathbf{d}_1(\pi,\pi^N)}{\sqrt{\epsilon}},$$

which implies

$$\|\hat{\pi}^{N,\epsilon} - \pi^\epsilon\|_{L^1(\omega)} \leq \frac{C\mathbf{d}_1(\pi,\pi^N)}{\sqrt{\epsilon}}.$$

Next, by considering $\text{Lip}(\psi) \leq 1$ which implies

$$\|\nabla\phi(0)\|_\infty \leq 1$$

we obtain

$$\mathbf{d}_1(\hat{\pi}^{N,\epsilon},\pi^\epsilon) \leq \mathbf{d}_1(\pi,\pi^N).$$

For estimate (45) we note that

$$\left|\hat{\pi}^{N,\epsilon}\log(\hat{\pi}^{N,\epsilon}) - \pi^\epsilon\log(\pi^\epsilon)\right| = \left|\int_0^1 (1 + \log(s\hat{\pi}^{N,\epsilon} + (1-s)\pi^\epsilon))ds(\hat{\pi}^{N,\epsilon} - \pi)\right|$$

$$\leq (1 + |\log(\delta)|)|\hat{\pi}^{N,\epsilon} - \pi^\epsilon|.$$

Therefore,

$$\|\hat{\pi}^{N,\epsilon}\log(\hat{\pi}^{N,\epsilon}) - \pi^\epsilon\log(\pi^\epsilon)\|_{L^1(\Omega)} \leq (1 + |\log(\delta)|)\|\hat{\pi}^{N,\epsilon} - \pi^\epsilon\|_{L^1} \leq C(1 + |\log(\delta)|)\frac{\mathbf{d}_1(\pi^N,\pi)}{\sqrt{\epsilon}}.$$

For estimate (46) we note that for any convex function $h$

$$h(x) - h(y) \geq \nabla h(y) \cdot (x - y) \implies h(y) - h(x) \leq \nabla h(y) \cdot (y - x).$$

Hence by applying the above to $h(x) = x\log(x)$ we obtain

$$\int \log(\eta^{N,\epsilon}(T))\eta^{N,\epsilon}(T) - \log\eta(T))\eta(T)dx \leq \int (1 + \log(\eta^{N,\epsilon}(T)))d(\eta^{N,\epsilon}(T) - \eta^\epsilon(T))$$

$$\leq \|1 + \log(\eta^{N,\epsilon}(T))\|_\infty\|\eta^{N,\epsilon}(T) - \eta^\epsilon(T)\|_{L^1}.$$

We now note that for any $g : R\mathbb{T}^d \to \mathbb{R}, \|g\|_\infty \leq 1$

$$\int g(x)d(\eta^{N,\epsilon} - \eta^\epsilon)(T) = \int \Gamma(T) \star gd(\hat{\pi}^{N,\epsilon} - \pi^\epsilon) \leq \|\nabla\Gamma(T) \star g\|_\infty\mathbf{d}_1(\hat{\pi}^{N,\epsilon},\pi^\epsilon)$$

$$\leq \frac{C}{\sqrt{T}}\mathbf{d}_1(\hat{\pi}^{N,\epsilon},\pi^\epsilon).$$

Taking the supremum yields

$$\|\eta^{N,\epsilon}(T) - \eta^\epsilon(T)\|_{L^1} \leq \frac{C}{\sqrt{T}}\mathbf{d}_1(\hat{\pi}^{N,\epsilon},\pi^\epsilon) \leq \mathbf{d}_1(\hat{\pi}^{N,\epsilon},\pi^\epsilon).$$

Finally, from Proposition D.3 above applied to $\rho = \eta^{N,\epsilon} - \frac{1}{\text{vol}(R\mathbb{T}^d)}$ we have

$$\eta^{N,\epsilon}(T,x) \geq -\|\rho(T)\|_\infty + \frac{1}{\text{vol}(R\mathbb{T}^d)} \geq \frac{1}{2}\frac{1}{\text{vol}(R\mathbb{T}^d)}$$

as well as

$$\eta^{N,\epsilon}(T,x) \leq \frac{1}{\text{vol}(R\mathbb{T}^d)} + \|\rho(T)\|_\infty \leq 2\frac{1}{\text{vol}(R\mathbb{T}^d)}$$

by our assumptions on $T > 0$. Therefore,

$$\|1 + \log(\eta^{N,\epsilon}(T))\|_\infty \leq C(1 + d\log(R))$$

and the result follows.

$\square$

**Remark B.4.** *It is worth noting that the empirical sample $\pi^N$ could have been regularized by any smooth kernel $\rho$. However the choice of the heat kernel which provides the same result as early stopping behaves quite nicely with respect to the metric $\mathbf{d}_1$ since it is a contraction. This is evident in Lemma B.3 above where we see that we do not pay any additional cost due to the mollification in (43).*

In fact the above techniques show the following general result regarding ESM to DSM bounds. Combining the above we can state the following general result, about ESM to DSM bounds.

**Theorem B.5.** *Let $\pi^1, \pi^2 \in \mathcal{P}(\Omega)$ denote two probability densities, such that*

$$\|\pi^i \log(\pi^i)\|_{L^1(\Omega)} < \infty \, \text{for } i = 1, 2 \tag{49}$$

*and $\delta > 0$ such that*

$$\pi^i(x) \geq \delta.$$

*Define $\rho^i : [0, T] \times \Omega \to \mathbb{R}$ as the solutions to*

$$\begin{cases} \partial_t \rho^i - \Delta \rho^i = 0 \text{ in } (0, T] \times \Omega, \\ \rho^i(0) = \pi^i \text{ in } \Omega. \end{cases} \tag{50}$$

*Then, up to a dimensional constant $C = C(d) > 0$*

$$\left| \mathcal{J}(\rho^2, \theta) - \mathcal{J}(\rho^1, \theta) \right| \leq C \Big( T \|\mathbf{s}_\theta\|_{C^2}^2 \mathbf{d}_1(\pi^1, \pi^2) + (1 + |\log(\delta)|) \|\pi^2 - \pi^1\|_{L^1} \Big). \tag{51}$$

## B.1   Proof of Theorem 3.3

Now we can prove Theorem 3.3.

*Proof.* From Proposition B.1 we obtain

$$\mathcal{J}(\eta^\epsilon, \mathbf{s}_{\theta*}) = \mathcal{J}_L(\eta^\epsilon, \mathbf{s}_{\theta*}) + 2\|\nabla\sqrt{\eta^\epsilon}\|_2^2$$

$$= \mathcal{J}(\eta^{N,\epsilon}, \mathbf{s}_{\theta*}) + 2\Big(\|\nabla\sqrt{\eta^\epsilon}\|_2^2 - \|\nabla\sqrt{\eta^{N,\epsilon}}\|_2^2\|\Big) + \Big(\mathcal{J}_L(\eta^\epsilon, \mathbf{s}_{\theta*}) - \mathcal{J}_L(\eta^{N,\epsilon}, \mathbf{s}_{\theta*})\Big).$$

By assumption we have the bound

$$\mathcal{J}(\eta^{N,\epsilon}, \mathbf{s}_{\theta*}) \leq e_{nn}.$$

By Proposition B.1

$$2\Big(\|\nabla\sqrt{\eta^\epsilon}\|_2^2 - \|\nabla\sqrt{\eta^{N,\epsilon}}\|_2^2\Big)$$

$$= \int_\Omega \pi^\epsilon(x) \log(\pi^\epsilon(x)) - \hat{\pi}^{N,\epsilon}(x) \log(\hat{\pi}^{N,\epsilon}(x)) dx - \int_\Omega \eta^\epsilon(T) \log(\eta^\epsilon(T)) - \eta^{N,\epsilon}(T) \log(\eta^{N,\epsilon}(T)) dx.$$

From Lemma B.3 we have that

$$\|\pi^\epsilon \log(\pi^\epsilon) - \hat{\pi}^{N,\epsilon} \log(\hat{\pi}^{N,\epsilon})\|_{L^1(\Omega)} \leq C(1 + |\log(\delta)|) \frac{\mathbf{d}_1(\pi^N, \pi)}{\sqrt{\epsilon}}.$$

Using the same argument and the fact that from maximum principle $\eta^\epsilon(T), \eta^{N,\epsilon} \geq \delta > 0$, we can show

$$\|\eta(T) \log(\eta(T)) - \eta^{N,\epsilon} \log(\eta^{N,\epsilon}(T))\|_{L^1(\Omega)} \leq (1 + |\log(\delta)|) \|\eta^\epsilon(T) - \eta^{N,\epsilon}(T)\|_{L^1(\Omega)}$$

$$\leq \frac{C}{\sqrt{T}} \mathbf{d}_1(\pi^\epsilon, \hat{\pi}^{N,\epsilon}) \leq \frac{C}{\sqrt{T}} \mathbf{d}_1(\pi, \pi^N).$$

Furthermore, by Proposition B.2 we have

$$\left| \mathcal{J}_L(\eta^\epsilon, \mathbf{s}_{\theta*}) - \mathcal{J}_L(\eta^{N,\epsilon}, \mathbf{s}_{\theta*}) \right| \leq CT \mathbf{d}_1(\hat{\pi}^{N,\epsilon}, \pi^\epsilon) \|\mathbf{s}_\theta\|_{C^2([0,T]\times\Omega)}^2 \leq CT \mathbf{d}_1(\pi^N, \pi) \|\mathbf{s}_\theta\|_{C^2([0,T]\times\Omega)}^2.$$

Putting everything together we have

$$\mathcal{J}(\eta, \mathbf{s}_{\theta*}) \leq e_{nn} + C\left(1 + \frac{|\log(\delta)|}{\sqrt{\epsilon}} + \frac{1}{\sqrt{T}} + T\|\mathbf{s}_\theta\|_{C^2([0,T]\times\Omega)}^2\right) \mathbf{d}_1(\pi^N, \pi).$$

$\square$

As an immediate Corollary we have the following.

**Corollary B.6.** *Using the same notation as in Section 3.5, if $e_{nn} > 0$ is such that*

$$0 \leq \mathcal{J}(\eta^{N,\epsilon}, \theta^*) < e_{nn} \tag{52}$$

*then*

$$0 \leq \mathcal{J}(\eta^{\pi^\epsilon}, \theta^*) < e'_{nn}, \tag{53}$$

*where for a dimensional constant $C = C(d) > 0$*

$$e'_{nn} = e_{nn} + C\Big(1 + \frac{|\log(\delta)|}{\sqrt{\epsilon}} + \frac{1}{\sqrt{T}} + T\|\mathbf{s}_\theta\|^2_{C^2([0,T]\times\Omega)}\Big)\mathbf{d}_1(\pi^N, \pi). \tag{54}$$

**Remark B.7** (On regularity implications of NN approximation)**.** *It is important to note that the norm of our NN approximation $\|\nabla \mathbf{s}_\theta\|_\infty$ is in fact dependent on $\epsilon > 0$ as we can see from the following*

$$\|\mathbf{s}_\theta\|_{C^1} \gtrsim \|\mathbf{s}_\theta\|_{L^2(\eta^{N,\epsilon})} \gtrsim \|\nabla \log(\eta^{N,\epsilon})\|_{L^2(\eta^{N,\epsilon})} - \sqrt{e_{nn}}$$

*and $\|\nabla \log(\eta^{N,\epsilon})\|_{L^2(\eta^{N,\epsilon})}$ blows up as $\epsilon \to 0$.*

We finish this subsection of technical results with an observation about the implications on the existence of a smooth NN-approximation.

**Proposition B.8** (ESM implies regularity)**.** *Let $\pi, \mathbf{s}_\theta$ and $e_{nn} > 0$ be as above for $\Omega = R\mathbb{T}^d$. Assume moreover, that $\|\nabla \mathbf{s}_\theta\|_\infty < \infty$. Then, $\pi = \pi(x)$ admits a density and in fact $\|\pi \log(\pi)\|_{L^1(\Omega)} < \infty$.*

*Proof.* Since $\|\nabla \mathbf{s}_\theta\|_\infty < \infty$, for some constant $C = C(R, d, e_{nn}) > 0$ it holds

$$2\|\nabla \sqrt{\eta^\pi}\|^2_{L^2} = \int_0^T \int |\nabla \log(\eta^\pi)|^2 d\eta^\pi(s, x) ds \leq C.$$

Moreover, by the regularizing properties of the diffusion we have

$$\int \eta^\pi(T) \log(\eta^\pi(T)) dx < \infty$$

and the result follows from Proposition B.1. $\qquad\qquad\qquad\qquad\qquad\qquad\qquad\qquad \square$

## C   Average DSM generalization

Using the previous results we can in fact show a result about the average error in Theorem B.5. The main observation here is that by Jensen's inequality when we take expectation with respect to the sample $\pi^N$ we no longer require a lower bound on our densities. However, as we will see we require a very restrictive assumption on the norm of our score function approximation

**Theorem C.1.** *(Average DSM generalization) Let $e_{nn}, A > 0$ and assume that for each sample $\hat{\pi}^N$ from $\pi$ there exists a $\mathbf{s}_{\theta^*}$ such that*

$$J(\eta^{N,\epsilon}, \theta^*) \leq e_{nn},$$

*with*

$$\|\mathbf{s}_{\theta^*}\|_{C^2} \leq A. \tag{55}$$

*Let $m_g(T)$ be the generated distribution from $\hat{\pi}^N$ (which is also random since $\hat{\pi}^N$ is random). Let $C > 0$ be the dimensional constant appearing in Proposition D.3 and $T \geq R^2$ large enough such that*

$$2CR^{-d}e^{-\frac{\omega T}{R^2}} \leq \frac{1}{2}\frac{1}{vol(R\mathbb{T}^d)},$$

*then*

$$\mathbf{E}\Big[\mathbf{d}_1(\pi, m_g(T))\Big] \leq CR^{\frac{3}{2}}(1 + \sqrt{A})(R^2 e^{-\frac{\omega T}{R^2}} + \sqrt{Te'_{nn}}),$$

*where*

$$e'_{nn} = \epsilon + e_{nn} + \frac{C}{N^{\frac{1}{2d}}}\Big(TRA^2 + \frac{1}{\sqrt{T}}(1 + d\log(R))\Big).$$

**Remark C.2.** *It is important to emphasize in the above that we are assuming that we may pick both an error $e_{nn} > 0$ and a bound $A > 0$ on the gradient of $\mathbf{s}_\theta$ globally for all random samples $\hat{\pi}^N$. Moreover, $\mathbf{s}_\theta$ is a weak approximation of $\nabla \log(\eta^{N,\epsilon})$ which although smooth for all $\epsilon > 0$ it does exhibit a blow-up as $\epsilon > 0$ tends to zero. Therefore, since by triangular inequality up to a constant $C = C(R, d) > 0$*

$$CA \geq \|\mathbf{s}_\theta\|_{C^2} \geq \|\mathbf{s}_\theta\|_{L^2(\eta^{N,\epsilon})} \geq \|D\sqrt{\eta^{N,\epsilon}}\|_{L^2} - e_{nn}$$

*the constant $A > 0$ will also exhibit a blow-up which can make the approximation worse. Therefore there could be a potential trade-off between NN approximation $e_{nn}$ becoming small and the value of the norm $A$.*

First we require a preliminary lemma.

**Lemma C.3.** *Under the same assumptions as in Theorem C.1 we have that*

$$\mathbf{E}\Big[\mathcal{J}(\eta^\epsilon, \mathbf{s}_\theta^N)\Big] \leq e_{nn} + \frac{C}{N^{\frac{1}{d}}}\Big(TA^2 + \frac{1}{\sqrt{T}}(1 + d\log(R))\Big). \tag{56}$$

*Proof.* We note that

$$\mathcal{J}(\eta^\epsilon, \mathbf{s}_{\theta^*}) = \mathcal{J}_L(\eta^\epsilon, \mathbf{s}_{\theta^*}) + 4\|D\sqrt{\eta^\epsilon}\|_2^2$$

$$= \mathcal{J}(\eta^{N,\epsilon}, \mathbf{s}_{\theta^*}) + 2\Big(\|D\sqrt{\eta^\epsilon}\|_2^2 - \|D\sqrt{\eta^{N,\epsilon}}\|_2^2\|\Big) + \Big(\mathcal{J}_L(\eta, \mathbf{s}_{\theta^*}) - \mathcal{J}_L(\eta^{N,\epsilon}, \mathbf{s}_{\theta^*})\Big).$$

By assumption we have the bound

$$\mathcal{J}(\eta^{N,\epsilon}, \mathbf{s}_{\theta^*}) \leq e_{nn}.$$

By Proposition B.1

$$2\Big(\|D\sqrt{\eta^\epsilon}\|_2^2 - \|D\sqrt{\eta^{N,\epsilon}}\|_2^2\Big)$$

$$= \int_\Omega \pi^\epsilon(x)\log(\pi^\epsilon(x)) - \hat{\pi}^{N,\epsilon}(x)\log(\hat{\pi}^{N,\epsilon}(x))dx - \int_\Omega \eta^\epsilon(T)\log(\eta^\epsilon(T)) - \eta^{N,\epsilon}\log(\eta^{N,\epsilon}(T))dx.$$

Taking expectations with respect to the sample $\hat{\pi}^N$ we get

$$\mathbf{E}\Big[\int_\Omega \pi^\epsilon(x)\log(\pi^\epsilon(x)) - \hat{\pi}^{N,\epsilon}\log(\hat{\pi}^{N,\epsilon})dx\Big] = \int_\Omega \pi^\epsilon(x)\log(\pi^\epsilon(x)) - \mathbf{E}\Big[\hat{\pi}^{N,\epsilon}(x)\log(\hat{\pi}^{N,\epsilon}(x))\Big]dx$$

$$\leq \int_\Omega \pi^\epsilon(x)\log(\pi^\epsilon(x)) - \mathbf{E}\Big[\hat{\pi}^{N,\epsilon}(x)\Big]\log(\mathbf{E}\Big[\hat{\pi}^{N,\epsilon}\Big])dx = 0.$$

In the above we used Jensen's inequality along with the fact that

$$\mathbf{E}\Big[\hat{\pi}^{N,\epsilon}\Big] = \pi^\epsilon.$$

Moreover, by Lemma B.3 we have that

$$\mathbf{E}\Big[\eta^{N,\epsilon}\log(\eta^{N,\epsilon}(T)) - \int_\Omega \eta^\epsilon(T)\log(\eta^\epsilon(T))dx\Big] \leq \frac{C}{\sqrt{T}}(1 + d\log(R))\mathbf{E}\Big[\mathbf{d}_1(\pi, \hat{\pi}^N)\Big]$$

$$\leq \frac{C}{\sqrt{T}N^{\frac{1}{d}}}(1 + d\log(R)).$$

Finally, from Proposition B.2 we have

$$\mathbf{E}\Big[\Big|\mathcal{J}_L(\eta, \mathbf{s}_{\theta^*}) - \mathcal{J}_L(\eta^{N,\epsilon}, \mathbf{s}_{\theta^*})\Big|\Big] \leq CT\mathbf{E}\Big[\|\mathbf{s}_\theta^N\|_{C^2}^2\mathbf{d}_1(\pi, \hat{\pi}^N)\Big] \leq \frac{CT}{N^{\frac{1}{d}}}A^2$$

Thus,

$$\mathbf{E}\Big[\mathcal{J}(\eta^\epsilon, \theta^*)\Big] \leq e_{nn} + \frac{C}{N^{\frac{1}{d}}}\Big(TA^2 + \frac{1}{\sqrt{T}}(1 + d\log(R))\Big),$$

and the result follows. □

*Proof.* By Theorem 3.1 we have that

$$\mathbf{d}_1(\pi, m_g(T)) \leq \sqrt{\epsilon} + \mathbf{d}_1(\pi^\epsilon, m_g(T)).$$

And

$$\mathbf{d}_1(m_g(T), \pi^\epsilon) = \mathbf{d}_1(m_g(T), m^\epsilon(T))$$

$$\leq CR^{\frac{3}{2}}(1 + \sqrt{\|Db^1\|_\infty})(\mathbf{d}_1(m^\epsilon(0), \frac{1}{\text{vol}(R\mathbb{T}^d)}) + \sqrt{T}\|\mathbf{b}^{\pi^\epsilon} - \mathbf{b}_{\theta^*}\|_{L^2(m^\epsilon)})$$

$$\leq CR^{\frac{3}{2}}(1 + \sqrt{A})\left(R^2 e^{-\frac{\omega T}{R^2}} + \sqrt{T\mathcal{J}(\eta^\epsilon, \theta^*)}\right).$$

Taking expectations and using the Lemma above yields the result. □

## D   Technical results

### D.1   HJB estimates

In this subsection, we gather all the regularity estimates for the HJB equations. The results are for the most part standard, we include however detailed proofs for the readers convenience. For further study into the techniques and topics we refer to [9, 10]. Although as mentioned in the introduction in the current work we assumed $\sigma(t) = \sqrt{2}$ and $f = 0$ in the OU processes (1), here we prove the regularity results for a general non-degenerate diffusion $\alpha(t)$. The addition of $f$ may be incorporated in the drift $b$.

**Proposition D.1.** *Let $\Omega \subset \mathbb{R}^d$, $v : \mathbb{R}^d \to \mathbb{R}$ and $b : [0, T] \times \Omega \to \mathbb{R}$ be given with $\|\nabla b\|_\infty < \infty$. Consider the solution $u : [0, T] \times \Omega \to \mathbb{R}$ of*

$$\begin{cases} -\partial_t u - \alpha(t)\Delta u + \frac{\alpha(t)}{2}|\nabla u|^2 + b \cdot \nabla u = 0 \text{ in } [0, T) \times \Omega, \\ u(T, x) = v(x) \text{ in } \Omega. \end{cases} \tag{57}$$

*Assume moreover that for some $M > 0$*

$$0 < \frac{1}{M} \leq \alpha(t).$$

*Then up to a universal constant $C > 0$ the following holds*

$$\|\nabla u(t)\|_\infty^2 \leq C(1 + \|\nabla b\|_\infty M)\|v\|_{C^1}. \tag{58}$$

$$(T - t)\|\nabla u(t)\|_\infty^2 \leq CM(T\|\nabla b\|_\infty + 1)\|v\|_\infty. \tag{59}$$

*Proof.* Let $z : [0, T] \times \Omega \to \mathbb{R}$ be given by

$$z = \frac{1}{2}|\nabla u|^2.$$

Differentiating the equation for $u$ in $i = 1, \cdots, d$ yields

$$-\partial_t u_i - \alpha(t)\Delta u_i + \alpha(t)\nabla u \cdot \nabla u_i + b \cdot \nabla u_i + b_i \cdot \nabla u = 0,$$

and thus multiplying by $u_i$ and summing over $i$, we obtain the following equation for $z$

$$\begin{cases} -\partial_t z - \alpha(t)\Delta z + \alpha(t)|\nabla^2 u|^2 + b \cdot \nabla z + \alpha(t)\nabla u \cdot \nabla z + \langle \nabla u \nabla b, \nabla u \rangle = 0, \\ z(T) = \frac{1}{2}|\nabla v|^2. \end{cases} \tag{60}$$

In the above $\nabla b$ is the matrix with entries

$$[\nabla b]_{i,j} = \frac{\partial b^i}{\partial x_j}$$

for

$$b = (b^1, \cdots, b^d).$$

Let $w : [0, T] \times \Omega \to \mathbb{R}$ be given by

$$w(t, x) = z - Cu$$

where $C > 0$ is a constant to be determined later. Assume that the function $w$ achieves its maximum at $(s_0, x_0) \in [0, T] \times \Omega$. We will look at the following cases:

- **Case 1:** Assume that $0 \leq s_0 < T$ and we will show that for $C > 0$ large enough this leads to a contradiction. At $(s_0, x_0)$ we have the optimality conditions

$$\nabla z(s_0, x_0) = C \nabla u(s_0, x_0)$$
$$\Delta z(s_0, x_0) \leq C \Delta u(s_0, x_0)$$
$$\partial_t z(s_0, x_0) \leq C \partial_t u(s_0, x_0).$$

Hence,

$$C(-\partial_t u(s_0, x_0) - \alpha(s_0) \Delta u(s_0, x_0) + b \cdot \nabla u(s_0, x_0))$$
$$\leq (-\partial_t z(s_0, x_0) - \alpha(s_0) \Delta z(s_0, x_0) + b \cdot \nabla z(s_0, x_0))$$

and using the respective equations for $u, z$ we obtain that at $(s_0, x_0)$

$$-C \frac{\alpha(s_0)}{2} |\nabla u|^2 \leq -\alpha(s_0) |\nabla^2 u|^2 - \langle \nabla u \cdot \nabla b, \nabla u \rangle - \alpha(s_0) \nabla u \cdot \nabla z.$$

Using the optimality conditions yields

$$\alpha(s_0) \frac{C}{2} |\nabla u|^2 + \alpha(s_0) |\nabla^2 u|^2 \leq -\langle \nabla u \cdot \nabla b, \nabla u \rangle \leq \|\nabla b\|_\infty |\nabla u|^2,$$

which is a contradiction if $C \geq 2\|\nabla b\|_\infty M$. Thus for

$$C_0 = 2\|\nabla b\|_\infty M$$

the function

$$z - C_0 u$$

achieves its maximum at some point $(T, x_0)$.

- **Case 2:** Assume that $s_0 = T$. Then, for every $(t, x) \in [0, T] \times \Omega$ we have

$$z(t, x) \leq w(T, x_0) + Cu(t, x) \leq \frac{1}{2} |\nabla v|^2 + C\|v\|_\infty + C\|u_+\|_\infty$$

and by maximum principle we also have that

$$\|u\|_\infty \leq \|v\|_\infty.$$

Thus, up to a universal constant

$$|\nabla u(t, x)|^2 \lesssim (1 + \|\nabla b\| M) \|v\|_{C^1},$$

which proves estimate (58).

Now we prove estimate (59). The proof follows the previous strategy only this time we consider the function
$$w(t, x) = (T - t)z - Cu,$$
where again $C > 0$ will be determined later. Assume that the maximum occurs at some point $(s_0, x_0)$.

1. **Case 1:** Assume that $s_0 < T$. Looking again the corresponding optimality conditions and using the equations for $u$ and $z$ we have at $(s_0, x_0)$

$$\left( \frac{C\alpha(s_0)}{2} - 1 \right) |\nabla u|^2 \leq -(T - s_0)\langle \nabla u \cdot \nabla b, \nabla u \rangle \leq T\|\nabla b\|_\infty |\nabla u|^2,$$

which is a contradiction if $C \geq 2TM\|\nabla b\|_\infty + 2M$.

2. **Case 2:** Assume that $s_0 = T$, then for all $(t, x) \in [0, T] \times \Omega$ we have

$$z(t, x) \leq w(T, x_0) + C_0 u(t, x) = -C_0 v(T, x) + C_0 u(t, x) \leq 2C_0 \|v\|_\infty.$$

Thus up to a dimensional constant we have

$$(T - t)|\nabla u|^2(t, x) \lesssim M(T\|\nabla b\|_\infty + 1)\|v\|_\infty.$$

$$\square$$

**Corollary D.2.** *Let $\Omega \subset \mathbb{R}^d$, $v : \mathbb{R}^d \to \mathbb{R}$ and $b : [0,T] \times \Omega \to \mathbb{R}$ be given with $\|\nabla b\|_\infty < \infty$. Given $\psi : \Omega \to \mathbb{R}$, with $\psi \geq 1$ consider the solution $\phi : [0,T] \times \Omega \to \mathbb{R}$ of*

$$\begin{cases} -\partial_t \phi - \alpha(t)\Delta\phi + b \cdot \nabla\phi = 0 \text{ in } [0,T) \times \Omega, \\ \phi(T,x) = \psi(x) \text{ in } \Omega. \end{cases} \tag{61}$$

*Assume moreover that for some $M > 0$*

$$0 < \frac{1}{M} \leq \alpha(t).$$

*Then up to a dimensional constant $C > 0$ we have the following estimates*

$$\|\nabla\phi(t)\|_\infty^2 \leq C\|\psi\|_{C^1}^3(1 + \|\nabla b\|_\infty M) \tag{62}$$

$$(T-t)\|\nabla\phi(t)\|_\infty^2 \leq C\|\psi\|_\infty^3 M(T\|\nabla b\|_\infty + 1) \tag{63}$$

*Proof.* Since $\psi \geq 1$ by the Maximum Principle we have that

$$1 \leq \phi(t,x) \leq \|\psi\|_\infty.$$

Thus we can define $u : [0,T] \times \Omega \to \mathbb{R}$ by the formula

$$\phi(t,x) = e^{-\frac{u(t,x)}{2}} \iff u(t,x) = -2\log(\phi(t,x)),$$

We then have

$$\partial_t\phi = -\frac{1}{2}\partial_t u\phi, \nabla\phi = -\frac{\nabla u}{2}\phi, \Delta\phi = \frac{|\nabla u|^2}{4}\phi - \frac{\Delta u}{2}\phi,$$

and so by substituting in the equation for $\phi$ we obtain

$$\begin{cases} -\partial_t u - \alpha(t)\Delta u + \frac{\alpha(t)}{2}|\nabla u|^2 + b \cdot \nabla u = 0, \\ u(T,x) = -2\log(\psi(x)). \end{cases} \tag{64}$$

The results will follow by Proposition D.1 for $v(x) = -2\log(\psi(x))$. First we need the following estimates

$$|v(x)| = 2|\log(\psi(x))| \leq 2\log(\|\psi\|_\infty) \leq 2\|\psi\|_\infty \text{ since } \psi \geq 1,$$

$$|\nabla v(x)| = 2\frac{|\nabla\psi(x)|}{|\psi(x)|} \leq 2\|\nabla\psi\|_\infty$$

$$\implies \|v\|_{C^1} \leq 2\|\psi\|_{C^1}.$$

Moreover, we note that

$$|\nabla u(t,x)|^2 = 4\frac{|\nabla\phi(t,x)|^2}{\phi^2} \geq 4\frac{|\nabla\phi(t,x)|^2}{\|\psi\|_\infty^2}.$$

Therefore, by Proposition D.1 estimate (58) we obtain that up to a dimensional constant $C > 0$

$$\|\nabla\phi(t)\|_\infty^2 \leq C\|\psi\|_\infty^2(1 + \|\nabla b\|_\infty M)\|\psi\|_{C^1} \leq C\|\psi\|_{C^1}^3(1 + \|\nabla b\|_\infty M).$$

Similarly by Proposition D.1, estimate (59) again up to a dimensional constant $C > 0$ we obtain

$$(T-t)\|\nabla\phi(t)\|_\infty^2 \leq C\|\psi\|_\infty^3 M(T\|\nabla b\|_\infty + 1).$$

$\square$

## D.2 Long time behavior of periodic heat equation.

The following result is classical and establishes exponential rate of convergence in $\mathbf{d}_1$ for solutions to the heat equation in the uniform distribution on the torus. We provide a proof for the readers convenience.

**Proposition D.3.** *Let* $\Omega = R\mathbb{T}^d$ *and* $m_i \in \mathcal{P}(\Omega), i = 1, 2$ *be two probability measures. Define* $\rho^i : [0, \infty) \times \Omega \to [0, \infty)$ *by*

$$\begin{cases} \partial_t \rho^i - \Delta\rho^i = 0 \text{ in } (0, \infty) \times \Omega, \\ \rho^i(0) = m_i \text{ in } \Omega. \end{cases}$$

*Then, there exists constants* $C = C(d) > 0, \omega = \omega(d) > 0$ *such that*

$$\mathbf{d}_1(\rho^2(t), \rho^1(t)) \le CRe^{-\frac{\omega t}{R^2}} \mathbf{d}_1(m_2, m_1) \text{ for } t \ge 0.$$

*In particular by choosing* $m_2 = \frac{1}{vol(R\mathbb{T}^d)}$

$$\mathbf{d}_1(\rho^1(t), \frac{1}{vol(R\mathbb{T}^d)}) \le CRe^{-\frac{\omega t}{R^2}} \mathbf{d}_1(m_1, \frac{1}{vol(R\mathbb{T}^d)}) \text{ for } t \ge 0. \tag{65}$$

*Proof.* First we assume that $R = 1$ and recall that if $\Gamma(t, x)$ denotes the heat kernel on the unit torus $\mathbb{T}^d$ then for a constant $C = C(d) > 0$ we have the normalizing effect

$$|\nabla\Gamma(t) \star g| \le C\frac{\|g\|_\infty}{\sqrt{t}}.$$

Fix $\psi : \mathbb{T}^d \to \mathbb{R}$ such that $\text{Lip}(\psi) \le 1$ and $\psi(0) = 0$. Such a function $\psi$ will also satisfy

$$\|\psi\|_\infty \le 1.$$

Fix $T > 0$ and let $\phi : \mathbb{T}^d \times [0, T] \to \mathbb{R}$ be given by

$$\begin{cases} -\partial_t\phi - \Delta\phi = 0 \text{ in } [0, T) \times \mathbb{T}^d, \\ \phi(T, x) = \psi(x). \end{cases} \tag{66}$$

By testing against $\phi$ in the equation satisfied by $\lambda = \rho^2 - \rho^1$ we obtain

$$\int \psi(x)d(\rho^2(T) - \rho^1(T))(x) = \int \phi(0)d(m_2 - m_1) \le \mathbf{d}_1(m_2, m_1)\|\nabla\phi(0)\|_\infty \le C\frac{\mathbf{d}_1(m_2, m_1)}{\sqrt{T}}.$$

Taking the supremum in $\psi$ yields

$$\mathbf{d}_1(\rho^2(T), \rho^1(T)) \le \frac{C}{\sqrt{T}}\mathbf{d}_1(\rho^2(0), \rho^1(0)).$$

Repeating the same argument as above with initial conditions $m_i = \rho^i(T)$, we obtain

$$\mathbf{d}_1(\rho^2(2T), \rho^1(2T)) \le \frac{C}{\sqrt{T}}\mathbf{d}_1(\rho^2(T), \rho^1(T)) \le \left(\frac{C}{\sqrt{T}}\right)^2\mathbf{d}_1(m_2, m_1)$$

and by induction for all $k \in \mathbb{N}$ we have

$$\mathbf{d}_1(\rho^2(kT), \rho^1(kT)) \le \left(\frac{C}{\sqrt{T}}\right)^k\mathbf{d}_1(m_2, m_1) = C^k T^{-\frac{k}{2}}\mathbf{d}_1(m_2, m_1).$$

Let $\theta \ge 0$ and fix a $k \in \mathbb{N}$ such that

$$kT \le \theta < (k + 1)T.$$

Since the heat operator defines a contraction in $\mathbf{d}_1$ we have that

$$\mathbf{d}_1(\rho^2(\theta), \rho^1(\theta)) \le \mathbf{d}_1(\rho^2(kT), \rho^1(kT)) \le (C\sqrt{T})^{-k}\mathbf{d}_1(m_2, m_1).$$

Moreover, we have that

$$-k \le -\frac{\theta}{T} + 1$$

thus

$$\mathbf{d}_1(\rho^2(\theta), \rho^1(\theta)) \le (C\sqrt{T})^{-\frac{\theta}{T}+1}\mathbf{d}_1(m_2, m_1)$$

and so choosing $T$ large enough so that

$$C\sqrt{T} \ge e$$

yields the result for $R = 1$. For a general $R > 0$, we simply apply the above to the functions $\lambda^i : [0, \infty) \times \mathbb{T}^d$ defined by

$$\lambda^i(t, x) = R^d \rho^i(R^2 t, Rx)$$

which yields

$$\mathbf{d}_1(\lambda^2(t), \lambda^1(t)) \leq C e^{-\omega t} \mathbf{d}_1(\lambda^2(0), \lambda^1(0)).$$

Since

$$\mathbf{d}_1(\lambda^2(t), \lambda^1(t)) = \sup_{\psi : \mathbb{T}^d \to \mathbb{R}, \mathrm{Lip}(\psi) \leq 1} \int_{\mathbb{T}^d} \psi(x) R^d (\rho^2(tR^2, Rx) - \rho^1(tR^2, Rx))$$

$$= \sup_{\psi} \int_{R\mathbb{T}^d} \psi\left(\frac{u}{R}\right) (\rho^2(tR^2, u) - \rho^1(tR^2, u))$$

$$= R \sup_{\psi : R\mathbb{T}^d \to \mathbb{R}, \mathrm{Lip}(\psi) \leq 1} \int_{R\mathbb{T}^d} \psi(u) (\rho^2(tR^2, u) - \rho^1(tR^2, u)) = R \mathbf{d}_1(\rho^2(R^2 t), \rho^1(R^2 t))$$

the result follows. □

