# OpenReview forum: "Score-based generative models are provably robust: an uncertainty quantification perspective"
_NeurIPS.cc/2024/Conference — NeurIPS 2024 poster_

### Official Review · Reviewer_DZY4 · 2024-07-11

**Soundness:** 3
**Presentation:** 2
**Contribution:** 3
**Rating:** 6
**Confidence:** 4

**Summary:**

This work studies the influence of different error terms for diffusion models from a continuous perspective under $W_1$ distance. They explain the reason why the early stopping parameter $\epsilon$ would lead to a memory phenomenon of diffusion models. To achieve these results, this work proposes a WUP theorem to explain the robustness of SGMs.

**Strengths:**

1.	The WUP theorem is novel since it can analyze generative flows instead of linear SDE and will arise independent interest.
2.	The analysis of the early stopping parameter can deepen the understanding of the memory phenomenon.
3.	The analysis of different objective functions is interesting.

**Weaknesses:**

1.	The abstract section mentions that stochasticity is the key mechanism ensuring SGM algorithms. However, this work does not discuss it in detail in the main content. It seems that the WUP theorem does not hold for the deterministic sampling process. It would be better to discuss it in detail.

**Questions:**

Please see the Weakness part.

Question 1: This work does not consider the influence of discretization error. It would be better to discuss the challenge when considering this error.

Question 2: For me, one interesting point is the balance of different terms when considering $\epsilon$. As shown in Theorem 3.3, $e_5$ term has $\sqrt{\epsilon}$ dependence and $e_2$ term has $1/\sqrt{\epsilon}$. It seems that there exists a balance between these terms when $N$ is finite. It would be better to discuss this balance in detail.

**Limitations:**

This work does not discuss the limitation and societal impact in a independent paragraph.

---

> ### Author Rebuttal · Authors · 2024-08-06
>
> **Response to Weakness #1**
>
> Thank you for pointing this out, we should have been more clear in our PDE terminology for the generative modeling community and will make sure to fix it. The *stochasticity* in the generative flow appears as the \emph{Laplacian operator} in our PDEs. Without the Laplacian, the Fokker-Planck equation is a continuity equation, which describes the density evolution of deterministic flows. There are two aspects of SGMs where stochasticity/Laplacian provides regularizing effects. The first is early stopping, i.e., adding some level of noise $\epsilon$ to the data distribution. This, in effect, runs the noising process for a short amount of time and immediately mollifies the initial distribution to have a smooth density.
>
> The second aspect is that the Laplacian is the key mechanism that provides the regularizing effects to the test function that then allows us to bound, for example, the stronger $\|\cdot\|_{L^1}$ normal by the weaker $\mathbf{d}_1$-norm. Without the Laplacian this effect would not be possible in general and in fact we would not have access to long time behavior results. We will make sure to add a detailed remark on this.
>
> **Response to Question #1**
>
> Yes, discretization error is indeed a source of error that should be looked into based on our analysis. This type of error can definitely be addressed by our framework; we however, defer this analysis for future work as it is a rather technical undertaking of its own. As we mentioned in Section 3.1, the most likely avenue for analysis is via the so-called modified equations [6]. That is, the numerical solution to the SDE is, in effect, the exact solution to a different *modified*  SDEs. The difference between the drift terms in the modified SDE and the approximated score can then be included when applying the WUP theorem. This further highlights the capabilities of the WUP perspective and our UQ angle.
>
> We refer the reviewer to our rebuttal summary we chose the errors we thought were most impactful and relevant to score-based generative modeling and PDE regularity theory. As raise by Reviewer gUwF Weakness \#1, the paper is already rather packed with a lot of new results and insights, and we believe adding the discretization error would diminish the readability and understanding of the main message of the paper.
>
> **Response to Question #2**
>
> This is a great point; in Remark 3.5 we briefly discuss how the bounds in Theorem 3.3 balances between the various sources of error or different properties of the data distribution. Moreover, it appears possible to optimize the bounds, which may further improve or inform the balance between the errors. This seems to be only possible as we work with integral probability metrics directly, which allows us to obtain sharper bounds than using a KL based approach.
>
> Moreover, roughly speaking our bounds (and all the previous bounds in the literature) obtain estimates on
>
> $d(m_{g},\pi)$
>
> where $m_g$ is the generated distribution and $\pi$ the true data distribution. However, we only have access to $\pi$ through the empirical distribution $\pi^N$, and we essentially estimate $d(m_g,\pi)$ through
>
> $d(m_g,\pi)\leq d(\pi,\pi^N)+d(\pi^N,\pi^{N, \epsilon})+d(\pi^{N,\epsilon} ,m_g)$
>
> where we further interpolate these distances through the addition of noise $\epsilon>0$. Notice that $d(\pi^N,\pi^{N,\epsilon})$ is finite when $d$ is an IPM, but not when it is a divergence.
>
> However, we believe it is currently rather premature to optimize the bounds as they are still qualitative for the most part, and are not fine-tuned (see our discussion to Reviewer PXMJ Weakness #1).  We will add a short discussion on how obtaining sharper bounds and then optimizing them in Remark 3.5 in a future version of the manuscript.
>
>
>
>
> [6] Abdulle, Assyr, et al. "High weak order methods for stochastic differential equations based on modified equations." 2012.

---

> > ### Comment · Reviewer_DZY4 · 2024-08-12
> >
> > Thanks for the detailed response. It would be better to add the discussion of W1 to the main content in the next version. I am satisfied with the answers and will maintain my positive score.

---

### Official Review · Reviewer_gUwF · 2024-07-11

**Soundness:** 3
**Presentation:** 2
**Contribution:** 3
**Rating:** 7
**Confidence:** 2

**Summary:**

The paper studies the robustness of score-based generative models (SGM) to different sources of error that are relevant in practice, such as the limited expressivity of the score function representation or the choice of reference distribution. Specifically, they upper bound the Wasserstein-1 and L1 distances between target and approximate distribution (given by an SGM) in terms of these different sources or error for both denoising and explicit score-matching objectives.

**Strengths:**

- Score-based generative models are extremely relevant within the machine learning community, and quantifying their uncertainty in terms of how well they capture the target distribution is an important research problem.
- The results are, to the best of my knowledge, new and it is impressive that they manage to isolate different sources of errors in their bounds while placing no restrictive assumptions on the target distribution.

**Weaknesses:**

- The main weakness of the paper is its dense presentation. The authors did a good job of motivating and describing their contributions in the introduction, but the rest of the paper is not easy to follow. I am not sure, there is much room for improvement within the limited space of 9 pages, but I would suggest including a discussion on actionable insights one can get from these bounds. Throughout the paper, the authors comment on how important robustness is for the reliability of generative models in general, and I agree, but I could not get an intuition about when one can expect these bounds to be tight so as to ensure the resulting model is trustworthy.
- While I respect the authors’ decision of going for an entirely theoretical paper, I cannot help but feel that some small scale experiments illustrating the tightness and usefulness of the bounds would be enlightening.

### Minor Issues
- Line 35: “contributes” should probably be plural here.
- Line 53: “recognizes” is misspelled.
- Line 61: no need for “of” after “study”.
- Line 100: if I’m not mistaken, the acronym SDE was not yet defined at this point in the text.
- Line 102: Albeit clear from the context, $W$ and $\eta$ have not yet been defined by this point in the text.
- Line 121: “however our results are generally apply”
- Line 331: Word missing after “used”, probably “for”.

**Questions:**

- In Section 6.2., the authors discuss an application of their bounds to likelihood-free inference. To that end, could the authors elaborate on how difficult it is to estimate their bounds in practice? Is accurately estimating the Lipschitz constant of the score function the main challenge?

**Limitations:**

This is mainly a theoretical work and I cannot foresee any direct societal impact stemming from this research.

---

> ### Author Rebuttal · Authors · 2024-08-06
>
> **Response to weakness #1**
>
> Thank you for the insightful feedback. As stated in our rebuttal summary, our goal is to introduce a PDE framework for error analysis of SGMs that are comparable to previous bounds, while being agnostic to situations where the data distribution is supported on a lower-dimensional manifold (see our response to reviewer PXMJ Weakness #1). To that end, the paper is organized to best understand the PDE framework for analyzing generative flows, i.e.
>
> 1. Study the evolution of test functions under the Kolmogorov backward equation of the true and approximate SDEs.
> 2. Bound the resulting integral probability metric using regularity estimates of the KBE.
> 3. Apply the resulting bound to the appropriate SGM setting.
>
> Therefore, the major actionable item the paper aims to convey is to consider PDE regularity theory for analyzing *any* generative flow—SGMs are just one choice.
>
> We will revise the paper to better highlight the important insights one may derive from our results. One reason we decided not to conjecture on actionable items is that our bounds are worst-case bounds that hold for any random sample of size $N$. While this shows that the bounds imply SGMs are robust, i.e., even in the worst case, the errors are bounded, we are not confident in the sharpness of the bounds.
>
> **Response to weakness #2**
> We appreciate the reviewer's point, and we believe the ultimate goal and impact of our approach is the ability to produce *a posteriori* bounds that can quantitatively capture the *confidence* a practitioner may have in a trained generative flow. In our rebuttal summary and the response to Reviewer PMXJ's Weakness #1, our primary objective is to showcase the connections between PDE theory and generative flows. Therefore, we make no claim in sharpness. In our response to the next question, we illustrate the usefulness of computing sharp bounds, which, however, requires further research. We defer numerics to future work, when the bounds can be sharpened and applied to more useful applications.
>
>
> **Response to Question #1**
> This is a great point, and it is an exciting topic we will study further in future work.
>
> We describe how one may think through how to compute the bounds. As you point out, the regularity bounds on the score function are critical. The Lipschitz constant of the score function will be dictated by the error on the ISM approximation we choose $e_{nn}$. To this end, a few factors need to be balanced.
>
> 1. If the underlying measure $\pi$ lies in a lower-dimensional manifold, its score function is undefined, since the term $\log(\pi)$ does not have meaning when $\pi$ is, for example, a Dirac mass. Therefore we require the addition of some noise with $\epsilon>0$, i.e., early stopping.
> 2. If $\epsilon >0$ is small and $e_{nn}>0$ is also chosen very small, then we are learning a potentially very rough function, and thus the Lipschitz norm will be large.
> 3. Moreover, we also have the regularizing effect of the Laplacian, which is highlighted in the choice of the test function. To bound stronger norms by weaker norms, we need more regularity on the score function we learn. For example, to bound the $\mathbf{d}_1$-norm by, say, some $H^{-s}$ norms, our bounds would require derivatives on the learned score function of order roughly $s>0$. The higher the $s$, the better the exponent on the size of the sample $N$, i.e., the rate. However, as pointed out in the previous item, the higher the $s$, the $ H^{-s}$-norm of the learned score function might also explode.
>
> With all of the above in mind, it seems that to answer your question, one would need a far more detailed analysis of the exact growth of these quantities, which is in fact the content of a work in progress. Finally, although we are not in a position yet to answer these questions, we believe that in the framework we introduced they can be posed as standard PDE questions.
>
> Once these questions are resolved, they may be most impactful for likelihood-free inference applications, e.g., [5]
>
> [5] Song, Yang, et al. "Solving inverse problems in medical imaging with score-based generative models." 2021.

---

> > ### Comment · Reviewer_gUwF · 2024-08-12
> >
> > Thanks for the detailed response. I am satisfied with the answers and will maintain my positive score.

---

### Official Review · Reviewer_PXMJ · 2024-07-13

**Soundness:** 3
**Presentation:** 3
**Contribution:** 3
**Rating:** 6
**Confidence:** 4

**Summary:**

This paper studies the generalization error of diffusion models. The major tool is the Wasserstein uncertainty propagation theorem. With such a result and the regularity analysis in PDE, the authors establish robust analysis for diffusion models with respect to various errors.

**Strengths:**

1. The authors examine the robustness of diffusion models from an uncertainty quantification perspective, that is not well-understood in prior work.
2. By leveraging the Wasserstein Uncertainty Propagation theorem, the authors provide the generalization error of diffusion models w.r.t. various error sources.
3. The paper is well-written and easy to follow. In particular, I appreciate the presentation of math derivation.

**Weaknesses:**

1. The first major concern is the connection and comparison to the literature. The bounds in Theorems 3.2 and 3.3 are not explicit. I suggest the authors provide clear sample complexity results w.r.t. problem parameters. Furthermore, the authors should discuss how the bounds improved the existing results in the literature.
2. The analysis in this paper heavily relies on the PDE theory and regularity analysis. I suggest the authors discuss relevant prior work in Section 1.2. Also, I suggest the authors briefly introduce UQ used in other ML problems beyond diffusion models. Furthermore, the score approximation and estimation theory should be mentioned as that is one source of the errors.

**Questions:**

I have some minor comments and questions:
1. The results established in this work are usually referred to sample complexity bounds or distribution estimation error. The authors use the name robust analysis while the meaning of "robustness" seems different from the one in robust optimization. I suggest the authors clarify this notion in the context.
2. The paper leverages the pathwise characterization of probability distribution generated by the forward process. I suggest the authors add more explanations and/or refer the readers to the literature when a PDE is introduced, e.g., Eq. (11). The same applies to all the FK and HJB equations.

---

> ### Author Rebuttal · Authors · 2024-08-06
>
> **Response to weakness #1**
> Yes you are correct that for the majority of our results, the bounds are more qualitative than quantitative. We refer the reviewer to our rebuttal summary, that our main goal  is to form a bridge between analysis of generative flows and PDE theory. We aim to illustrate our methods' strengths with specific applications such as the ESM and DSM bounds which, as you have highlighted, are not sharp. Establishing sharp constant bounds is important and beyond the scope of this work. See our response to reviewer gUwF about how we may investigate computing the bounds. To address your concerns, we highlight our novel quantitative and qualitative contributions:
> - **Qualitative:**
>   - ESM Bound (Theorem 3.2) and DSM bound (Theorem 3.3) show the following improvements:
>     1. We obtain estimates in norms better than the KL divergence and, in fact, are able to bound the stronger total variation ($\|\cdot\|_{L^1}$ norm) by the weaker Wasserstein-1 ($\mathbf{d}_1$) norm.
>     2. We obtain estimates for the $\mathbf{d}_1$ norm *without* bounding the KL divergence and applying Pinsker's inequality.
>     3. Theorems B.5 and C.1 show it is possible to relate an *a priori* unknown error in the ESM objective with the error from the DSM objective (at least on average).
>
>   Our results are *agnostic* to the manifold hypothesis, i.e., they apply both in the case when the distribution is degenerate, and when it admits a density. We highlight this fact throughout the paper (Sections 1.1, 1.2, and Theorem 3.3). Estimate (12) in Theorem 3.2 exists in previous literature only under the assumption that the data distribution is absolutely continuous with respect to a Gaussian [1,2,3]
>
> - **Quantitative:**
>  While we track the major quantities in our Theorems, you are correct that there are important constants which are not explicit. In particular using the same notation as in our Theorems we use:
>   1. A constant $C>0$ which depends only on the dimension.
>   2. A constant $\omega$ related to the exponential rate of convergence to the stationary measure in the heat equation.
>   3. A constant $\delta>0$ which captures the lower bound a measure obtains when we apply some diffusion of level $\epsilon>0$.
>
> $C$ and $\omega$ are related to classical problems in PDEs and should be computable in the future when our framework is further refined. We chose to focus on the core connections between generative modeling and PDE theory, as explicitly computing these constant would make the paper far too technical in PDE theory.
>
> A major focus of future work is to compute these constants more explicitly. For example, it is important to understand if the dependence of $C>0$ on the dimension is linear, polynomial or something worse. For $\delta$, this appears to be more difficult to compute and would require more assumptions on the underlying measure $\pi$. While adding some noise $\epsilon>0$ (at least on the torus) guarantees that the measure will gain support everywhere the exact size is a challenging problem.
>
> **Response to weakness #2**
> Thank you for your feedback. We will include discussion about relevant aspects of PDE theory that are most useful in our work. Moreover, while the use of generative modeling for UQ has been explored, to our knowledge, the UQ perspective for studying generative modeling is uncommon. We highlight [4] which derives a type of Wasserstein Uncertainty Propagation bound for the W2 distance, although they do not refer to their work as a UQ perspective. We discuss this work in Section 1.2. Regarding score approximation and estimation theory, we describe this error as source #3, score expressivity, in Section 3.1. We will rephrase this source of error in the next version of the manuscript.
>
> **Response to Question #1**
> Thank you for your comments. There are multiple senses of robustness we use in our work. We will clarify these notions further in a future version of the manuscript. A subtle but key result of our work is that our sample complexity bounds are *worst case* bounds for *any* random sample of size $N$. As the worst case bound is finite and controlled, it demonstrates that score-based generative modeling is robust. See the discussion after Theorem 3.3. This sense of robustness is actually similar that of robust optimization, where the method produces the correct value for worst case choices of the parameters or initial conditions.
>
> In contrast to previous analysis, our results are *agnostic* to the manifold hypothesis. Past work assumes the data distribution has a density in $\mathbb{R}^d$, i.e., not supported on a low-dimensional manifold. As our results are independent of the manifold hypothesis, it explains why SGMs are robust, i.e., work well, even when the data distribution is degenerate. The regularizing properties of the Laplacian are what enable this robustness.
>
> **Response to Question #2**
> Thank for for reminding us to include appropriate references to PDE theory for the generative modeling community. We will be sure to include references that will help both the PDE and generative modeling communities more easily explore each other's previous work. One goal of this paper is to showcase the connections between the two communities and we hope it may initiate such collaborations. We will add important sources, and provide some extra discussion to guide the reader.
>
> [1] Lee, Holden, Jianfeng Lu, and Yixin Tan. "Convergence of score-based generative modeling for general data distributions." 2023.
>
> [2] Chen, Sitan, et al. "Sampling is as easy as learning the score: theory for diffusion models with minimal data assumptions." 2022.
>
> [3] Chen, Hongrui, Holden Lee, and Jianfeng Lu. "Improved analysis of score-based generative modeling: User-friendly bounds under minimal smoothness assumptions." 2023.
>
> [4] Kwon, Dohyun, Ying Fan, and Kangwook Lee. "Score-based generative modeling secretly minimizes the wasserstein distance." 2024.

---

> > ### Comment · Reviewer_PXMJ · 2024-08-12
> >
> > Thank you for your detailed response. I have raised the score.

---

### Author Rebuttal · Authors · 2024-08-06

We thank all the reviewers for their careful reading, time, and insightful comments on our work. They will be invaluable for improving the current manuscript and for future work.

We emphasize the organizing principle behind our paper. **The primary goal of our work is to establish connections between PDE theory and flow-based generative models.  We create a proper framework where various sample complexity and error bounds can be the derived from existing PDE stability results and regularity estimates.** In particular, the current paper employs PDE regularity theory and the regularizing properties of the Fokker-Planck equation to analyze score-based generative models. Moreover, the Wasserstein Uncertainty Propagation theorem we derive is motivated by the model-form uncertainty quantification problem that naturally arises in score-based generative models.  Many presentation choices and trade-offs in our work are made with the organizing principle in mind.

---

### Decision · Program_Chairs · 2024-09-25

**Decision:**

Accept (poster)

**Comment:**

This paper gives theoretical analysis of score-based generative models from an uncertainty quantification perspective, showing the impact of errors from different sources. They use a Wasserstein uncertainty propagation theorem using PDE theory, which is potentially of broader interest beyond SGM's. Reviewers received the paper positively and thought it was generally well-written and mathematically sound, with solid technical contributions, though noted a lack of tightness in the bounds (including some unspecified constants) and discretization analysis. I recommend acceptance, with the reviewers' suggestions to improve clarity of presentation.